# Non defect-stabilized thermally stable single-atom catalyst

Rui Lang[1], Wei Xi[2], Jin-Cheng Liu [3], Yi-Tao Cui [4], Tianbo Li[1,5], Adam Fraser Lee [6], Fang Chen[1], Yang Chen[1,5], Lei Li[7], Lin Li[1], Jian Lin[1], Shu Miao[1], Xiaoyan Liu[1], Ai-Qin Wang[1], Xiaodong Wang[1], Jun Luo[2], Botao Qiao [1,8], Jun Li[3,9] & Tao Zhang[1]

Surface-supported isolated atoms in single-atom catalysts (SACs) are usually stabilized by diverse defects. The fabrication of high-metal-loading and thermally stable SACs remains a formidable challenge due to the difficulty of creating high densities of underpinning stable defects. Here we report that isolated Pt atoms can be stabilized through a strong covalent metal-support interaction (CMSI) that is not associated with support defects, yielding a high-loading and thermally stable SAC by trapping either the already deposited Pt atoms or the $PtO_2$ units vaporized from nanoparticles during high-temperature calcination. Experimental and computational modeling studies reveal that iron oxide reducibility is crucial to anchor isolated Pt atoms. The resulting high concentrations of single atoms enable specific activities far exceeding those of conventional nanoparticle catalysts. This non defect-stabilization strategy can be extended to non-reducible supports by simply doping with iron oxide, thus paving a new way for constructing high-loading SACs for diverse industrially important catalytic reactions.

[1] State Key Laboratory of Catalysis, Dalian Institute of Chemical Physics, Chinese Academy of Sciences, Dalian 116023, China. [2] Center for Electron Microscopy and Tianjin Key Lab of Advanced Functional Porous Materials, Institute for New Energy Materials, School of Materials, Tianjin University of Technology, Tianjin 300384, China. [3] Department of Chemistry & Key Laboratory of Organic Optoelectronics and Molecular Engineering of the Ministry of Education, Tsinghua University, Beijing 100084, China. [4] Synchrotron Radiation Laboratory, Laser and Synchrotron Research Center (LASOR), The Institute for Solid State Physics, The University of Tokyo, 1-490-2 Kouto, Shingu-cho Tatsuno, Hyogo 679-5165, Japan. [5] University of Chinese Academy of Sciences, Beijing 100049, China. [6] School of Science, Royal Melbourne Institute of Technology University, Melbourne VIC3001, Australia. [7] Synchrotron Radiation Nanotechnology Center, University of Hyogo, 1-490-2 Kouto, Shingu-cho Tatsuno, Hyogo 679-5165, Japan. [8] Dalian National Laboratory for Clean Energy, Dalian 116023, China. [9] Department of Chemistry, Southern University of Science and Technology, Shenzhen 518055, China. These authors contributed equally: Rui Lang, Wei Xi and Jin-Cheng Liu. Correspondence and requests for materials should be addressed to J.L. (email: jluo@tjut.edu.cn) or to B.Q. (email: bqiao@dicp.ac.cn) or to J.L. (email: junli@tsinghua.edu.cn)

Heterogeneous catalysis is pivotal to the modern chemical industry[1], with many heterogeneous catalysts comprising transition metals deposited over a solid support phase[2]. A recent evolution in this field is the dispersion of isolated, individual metal atoms over the support to form single-atom catalysts (SACs)[3]. Such SACs have attracted considerable attention as a new frontier in heterogeneous catalysis[4,5], affording enhanced precious metal thrifting[6] and atom economy[7–9], improved active site homogeneity[10–12], and the ability to tune the metal-support interface with unprecedented control[13–16], resulting in superior catalytic activity[17–20] and/or high selectivity[21–23]. A diverse range of supports are known to stabilize isolated metal atoms, notably through pinning at electronic and/or structural defects associated with coordinatively unsaturated sites (CUS) by experimental and density functional theory (DFT) studies (Supplementary Appendices I and II)[24–33]. The concentration of atomically-dispersed metal atoms, and their stability, are therefore intrinsically linked to the density and stability of defects as described in recent reviews (Supplementary Appendix III)[34,35]. However, it is extremely difficult to create a high density of thermally stable support defects[36], and hence the fabrication of high metal loading SACs remains currently a formidable challenge.

Here we report the discovery that single metal atoms can be stabilized through a strong covalent metal-support interaction (CMSI) that is not associated with surface defects, enabling the genesis of high concentrations of thermally stable single atoms, even on low-surface-area materials. In situ genesis of ferric oxide supported Pt SAC from Pt nanoparticles (NPs) is verified in methane combustion reaction, one of the primary means of human energy production and important for mitigating environmental challenge associated with $CH_4$ emission. Experimental and DFT studies demonstrate that metal oxide reducibility dictates the ability of a support to anchor isolated Pt atoms: $\alpha$-$Fe_2O_3$ favors atomically dispersed Pt, whereas $Al_2O_3$ favors NP sintering. However, the non defect-stabilization strategy can be extended to non-reducible oxide by simply doping with iron oxide. This finding provides a promising method to fabricate high-metal-loading SACs with excellent thermal stability.

## Results

### Co-precipitation prepared thermally stable $Pt_1/FeO_x$.

A 1.8 wt% $Pt/FeO_x$ catalyst was first prepared by co-precipitation (the synthesis details are in the Methods section), denoted as $Pt_1/FeO_x$, and subsequently calcined at 800 °C in air for 5 h, denoted as $Pt_1/FeO_x$-C800. Aberration-corrected high-angle annular dark-field scanning transmission electron microscopy (AC-HAADF-STEM) with sub-angstrom resolution was used to compare the nature of platinum species between these two materials. In neither case were Pt nanoclusters or NPs observed (Supplementary Figures 1 and 2). $Pt_1/FeO_x$ comprised isolated Pt atoms dispersed on the $FeO_x$ support with some Pt atoms exactly aligned with the Fe atomic columns (Fig. 1a–b). Corresponding CO diffuse reflectance infrared Fourier transform spectroscopy (DRIFTS) of $Pt_1/FeO_x$ (Supplementary Figure 1c) shows a coverage-independent band at ~2089 $cm^{-1}$ and the absence of bridged adsorption of CO band (~1860 $cm^{-1}$), which is consistent with the CO linearly bound to isolated Pt atoms[3]. Note that our CO infrared bands are broader than those recently reported by Christopher and co-workers[37]. However, this is unsurprising because in their work ultra-dilute (0.05 wt%) Pt loadings were used to immobilize only one Pt atom per 5 nm monodispersed anatase nanocrystal. Such low loadings favor the population of only the most reactive titania surface sites, and hence a homogeneous local environment. Platinum sintering was not observed following 800 °C calcination by either AC-HAADF-STEM

(Fig. 1c–d and Supplementary Figure 2a–b), or powder X-ray diffraction (Supplementary Figure 2c) analysis of the $Pt_1/FeO_x$-C800 sample. X-ray absorption spectroscopy (XAS) studies provide further evidence for the complete dispersion of Pt atoms. Figure 1e presents the Fourier transform radial distribution functions of $k^3$-weight extended X-ray absorption fine structure (EXAFS) spectra for $Pt_1/FeO_x$ before and after calcination. Similar to our previous reports[3], no Pt–Pt bond contribution was found. This observation is in marked contrast to the significant sintering typically noted for small metal species/atoms sintered to calcination at elevated temperature[38,39], and indicates a very strong interaction between Pt atoms and $Fe_2O_3$ that inhibits migration and sintering of the former. X-ray absorption near edge structure (XANES) measurements confirmed that the chemical state of Pt atoms before and after calcination resembled that $PtO_2$ with tetravalent Pt(IV) (Supplementary Figure 2d) rather than metallic platinum with zerovalent Pt(0)[3].

### Pt NP-generated high loading Pt SACs on $Fe_2O_3$ support.

Pt NPs are known to liberate mobile $PtO_2$ species at elevated temperature under an oxidizing atmosphere[40], which may be subsequently trapped by either larger Pt NPs (Ostwald ripening) or unique oxide sites or crystalline facets resulting in metal dispersion[41–45]. In the present case, wherein a strong interaction between Pt atoms and the $Fe_2O_3$ support is apparent, we anticipate that Pt atoms vaporized from a NP could be trapped and stabilized by the $Fe_2O_3$ surface. To test this hypothesis, pre-formed colloidal Pt NPs were supported on $Fe_2O_3$ at a 0.3 wt% loading and the ethylene glycol stabilizer was removed by a 500 °C calcination (denoted as $0.3Pt/Fe_2O_3$-NP), prior to various high-temperature calcination treatments. HAADF-STEM imaging revealed the presence of 2–3 nm diameter Pt NPs (Supplementary Figure 3a–b). In contrast to $Pt_1/FeO_x$ materials, CO DRIFTS of $0.3Pt/Fe_2O_3$-NP exhibited both bridging (1839 $cm^{-1}$) and coverage dependent linear bound CO (2072–2064 $cm^{-1}$) (Supplementary Figure 3c), which is consistent with extended arrays of Pt atoms. Higher temperature calcination (800 °C, 5 h) resulted in the complete disappearance of these Pt NPs (denoted as $0.3Pt/Fe_2O_3$-C800, Supplementary Figure 4a), and concomitant formation of a very high density of isolated Pt atoms (Supplementary Figure 4b). Fig. 2 presents the Fourier transform radial distribution function of the Pt $L_{III}$-edge $k^3$-weighted EXAFS spectrum of $0.3Pt/Fe_2O_3$-C800, alongside reference distribution functions for Pt foil and $PtO_2$ (corresponding EXAFS spectra and fits appear in Supplementary Figure 5 and Supplementary Table 1). Two prominent scattering distances are observed for $0.3Pt/Fe_2O_3$-C800 at ~1.7 Å and 2.7 Å attributed to Pt–O and Pt–Fe contributions[3,27], respectively, confirming only the sole presence of atomically-dispersed Pt atomic species.

Note that some Pt NPs remained visible by electron microscopy after a lower temperature (600 °C) calcination, albeit more loosely coordinated to the underlying support (Supplementary Figure 4c). An oxidizing atmosphere was critical to observe this high-temperature dispersion; annealing $0.3Pt/Fe_2O_3$-NP under Ar at 800 °C promoted Pt aggregation into larger 3–5 nm particles (Supplementary Figure 4d). This observation strongly implicates oxidized Pt species in Pt NP dispersion (Fig. 3), and indeed X-ray photoelectron spectra (XPS) (Supplementary Figure 6c) provide direct evidence for high-valent (near + IV) Pt in the $0.3Pt/Fe_2O_3$-C800 catalyst. This is quite different from those divalent (+II) Pt ions dominating on $CeO_2$ support after high-temperature calciantion[46,47], and hence further evidences a stronger Pt-$Fe_2O_3$ interaction that partially arises from covalent bonding interaction[48,49]. Three requirements emerge for the successful conversion of Pt NPs into atomically dispersed Pt: a

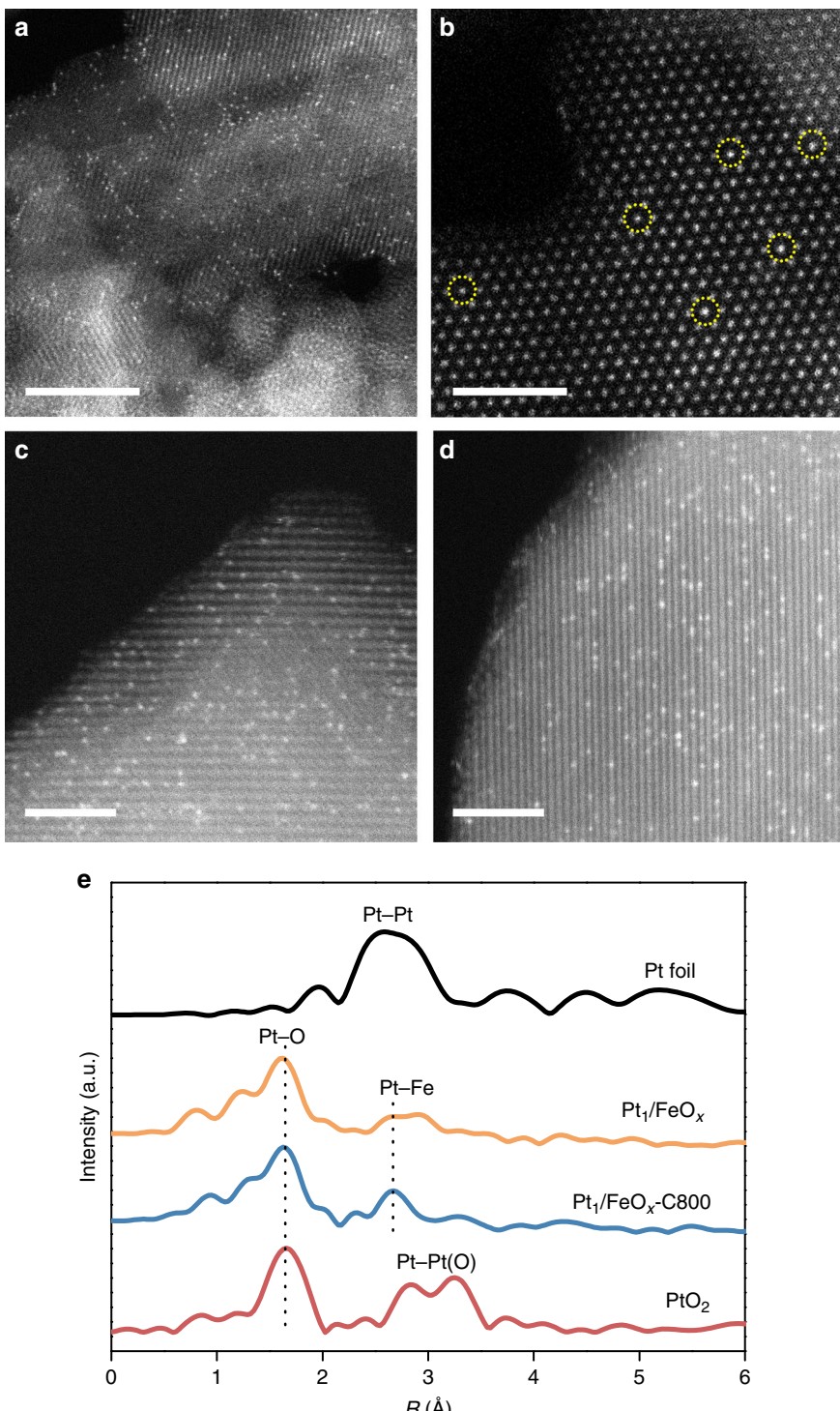

**Fig. 1** Structure of Pt$_1$/FeO$_x$. **a**, **b** AC-HAADF-STEM images of Pt$_1$/FeO$_x$, and **c**, **d** Pt$_1$/FeO$_x$-C800, highlighting atomically dispersed Pt (circled in **b**). 5 nm scale bar for panel **a**, and 2 nm scale bars for **b**, **c**, **d**. **e** Fourier transform radial distribution function of the Pt L$_{III}$-edge $k^3$-weighted EXAFS spectra of Pt$_1$/ FeO$_x$ before and after calcination in comparison with PtO$_2$ and Pt foil

high temperature to promote Pt mobility; the presence of molecular O$_2$ to partially oxidize the surface of Pt NPs, thereby enabling vaporization of (mobile) PtO$_2$; and a strong interaction between support and surface Pt atoms. Consequently, highly stable static single atoms on heterogeneous surface can be fabricated[50].

The first condition is met for H$_2$PtCl$_6$ functionalized α-Al$_2$O$_3$ (Supplementary Figure 7a–b) subjected to the same high-temperature processing. However, in this case the weak metal-

support interaction favors Ostwald ripening and the growth of large (10~50 nm) Pt particles (Supplementary Figure 7c–d) over their dispersion, as previously reported[42]. Since the particle size in the fresh sample ranges from 2–20 nm (Supplementary Figure 7b), and the dynamics of Pt NP dispersion into Pt atoms are related to particle size[51], a control experiment was further performed, in which 2–3 nm Pt NPs were deposited over Al$_2$O$_3$ (Supplementary Figure 7e–f). Sintering was again observed after heat treatment (Supplementary Figure 7g–h), confirming a weak interaction

between alumina and Pt species. We note that dispersion of < 1 nm Pt NPs was recently reported over MCM-22[51], which may reflect restricted migration of Pt species through the microporous zeolite network.

Having identified conditions under which Pt NPs can be dispersed over an $Fe_2O_3$ surface, we explored the effect of Pt loading with a view to maximizing the density of isolated atoms achievable, and thereby compensate for the extremely low surface area (only $5$–$10\ m^2\ g^{-1}$) of the $Fe_2O_3$ support after the requisite 800 °C calcination. A higher 1 wt% loading of the same 2–3 nm Pt NPs (denoted as $1Pt/Fe_2O_3$-NP) were also fully transformed into isolated atoms by calcination at 800 °C (Supplementary Figure 8a–b). Low energy ion scattering (LEIS) evidenced a clear decrease in the Pt:Fe atomic ratio in the outermost surface layer upon calcining the stabilizer free $1Pt/Fe_2O_3$-NP to 800 °C (Supplementary Figure 9). Since the total Pt loading determined by inductively coupled plasma spectrometry-atomic emission spectrometry (ICP-AES) remained unchanged, it appears that

some Pt atoms diffuses into the near sub-surface region of the $Fe_2O_3$ support, possibly by a cation-exchange process[52,53]. However, a further increase in Pt loading to 2 wt% resulted in the formation of Pt clusters alongside isolated atoms (Supplementary Figure 8c–d), in line with the theoretical maximum loading (~1.5 wt% for $Fe_2O_3$ possessing a surface area of $10\ m^2\ g^{-1}$, see Methods). A high density of clusters and few single isolated atoms were observed for a 4.5 wt% Pt loading (Supplementary Figure 8e–f). The maximum concentration of atomically-dispersed Pt loading is clearly dictated by the very low support surface area. The emergence of Pt clusters for loadings ≥2 wt% confirms that any Pt atoms diffusing into the support must be confined to the immediate subsurface region and not migrated into the bulk $Fe_2O_3$ lattice, since the latter pathway would enable isolated atoms to remain the dominant surface species at far higher Pt loadings.

Disintegration of Pt NPs during high-temperature calcination was directly visualized by in situ HAADF-STEM, and one $Fe_2O_3$ crystal of $1Pt/Fe_2O_3$-NP sample was randomly picked to observe the process. Prior to calcination, about 300 Pt particles were visible across the $Fe_2O_3$ support, and the average particle size was about 3 nm (Fig. 4a–c). However, after heating to 800 °C under a flow of 1 bar $O_2$ for 20 min, the total number of Pt NPs decreased to ~200, since many of the 2–3 nm Pt NPs shrank and/or disappear entirely, while a few larger particles (size above 10 nm) remained (Fig. 4b–c versus Fig. 4e–f). The loss of small clusters was confirmed by higher resolution, time-resolved imaging (Fig. 4g–i taken from the Supplementary Movie 1), which reveal smaller Pt clusters (e.g., particle 1) disintegrated in less than 35 s, while larger Pt clusters (e.g., particle 2) vanished in 45 s. Previous reports suggest that smaller metal crystallites disperse as adatoms, which migrate towards neighboring (larger) particles and eventually coalesce at elevated temperature via Ostwald Ripening, driven by lowering of the total surface energy[54]. In this case, however, particle disappearance occurs in the absence of such coalescence, consistent with the accompanying genesis of atomically-dispersed Pt entities.

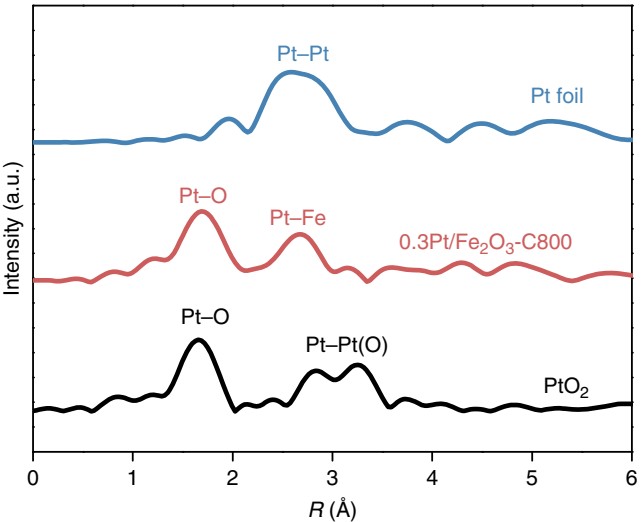

**Fig. 2** X-ray absorption spectroscopy of the $0.3Pt/Fe_2O_3$-C800 catalyst. Fourier transform Pt $L_{III}$-edge radial distribution functions of the $0.3Pt/Fe_2O_3$-C800 catalyst in comparison with $PtO_2$ and Pt foil

**DFT Studies**. The preceding electron microscopy, X-ray spectroscopy and diffraction, and CO chemisorption measurements provide both clear evidence of, and insight into the driving force for, the formation of atomically-dispersed Pt following high-

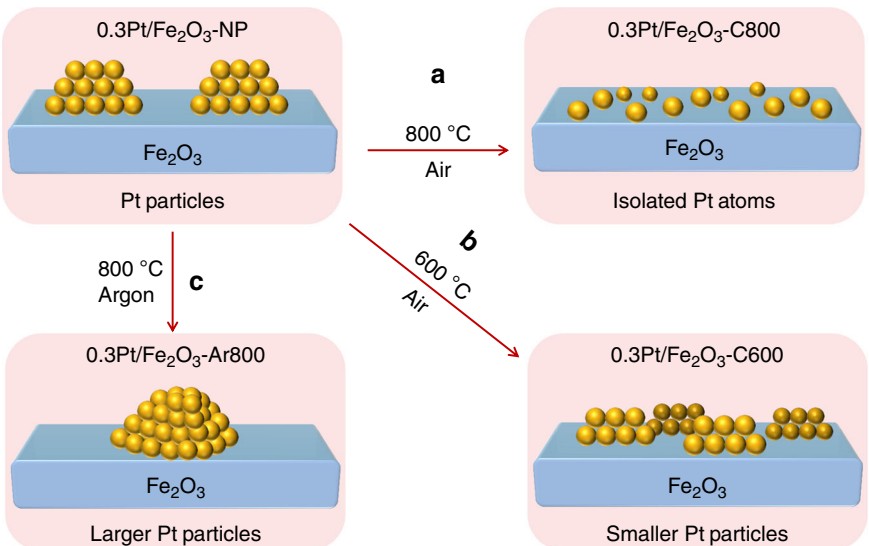

**Fig. 3** Illustration of thermally induced Pt nanoparticle restructuring. **a**, **b** Calcination under oxygen, or under an inert atmosphere (**c**), resulting in dispersion as single-atoms or particle sintering, respectively

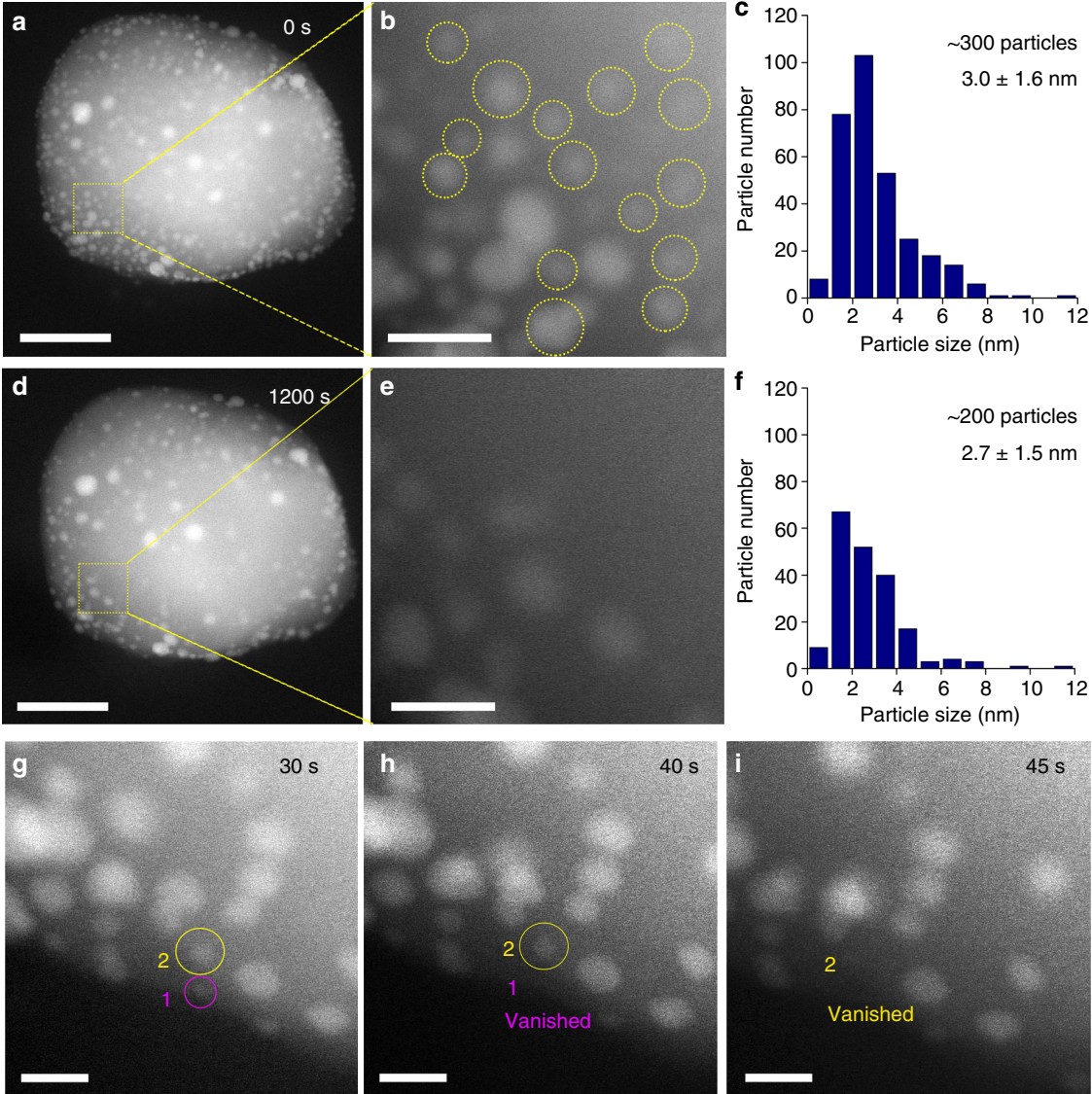

**Fig. 4** In situ characterization of Pt NP oxidative dispersion. **a–c** HAADF-STEM images and size distribution histogram of 1Pt/Fe$_2$O$_3$-NP before, and **d–f** after in situ calcination at 800 °C under 1 bar flowing O$_2$ for 20 min: the yellow squares in panels a and d show the same sample area. A 50 nm scale bars in **a**, **d** and 10 nm scale bars in **b**, **e**. Yellow circles in panel b highlighted the particles missing after calcination for 20 min. **g–i** Sequential HAADF-STEM images from the same area showing the dissociation of small particles (labeled by purple and yellow circles) during in situ calcination: 5 nm scale bars; elapsed time (in seconds) indicated in upper right corner of each image

temperature calcination; NP dispersion is triggered by the formation of PtO$_2$ (Supplementary Figure 6). Computational modeling via DFT calculations (Fig. 5a–b) suggests that the evaporation free energy of PtO$_2$ (the dominant surface species under an oxidizing environment) from a Pt(221) step (model in Supplementary Figure 10) is about −0.61 eV per PtO$_2$ at 800 °C, whereas the evaporation energy of Pt$_1$ (the dominant species under inert environments, Supplementary Figure 4d) can reach as high as 4.00 eV, and hence is energetically strongly disfavoured due to CMSI of Pt$_1$ to the surface atoms. By comparison, PtO$_2$ evaporation at 600 °C is virtually thermo-neutral or slightly endothermic (0.07 eV), indicating the critical role of temperature in dispersing Pt NPs. This result is consistent well with the experimental results that the dispersion of NPs only occurred with calcination >600 °C (Fig. 3, Supplementary Figures 4a–c).

This temperature-dependent evolution of PtO$_2$ dissociation energetics from a Pt(221) step (representative of the partially oxidized surface of Pt NPs following high-temperature

calcination) mirrors that of the catalytic activity for methane oxidation, as discussed below. The dissociative adsorption of PtO$_2$ on an oxygen terminated Fe$_2$O$_3$(0001) surface (and concomitant desorption of one O$_2$ molecule) is highly exothermic at 800 °C (calculated as −2.46 eV, Fig. 5c–d), with PtO$_2$ dissociative capture even more favorable at lower temperature. Note that the evaporation, migration, and trapping do not require the presence of anion or cation vacancies on the support: no Lewis acid sites were detected on the Fe$_2$O$_3$ support by NH$_3$ temperature-programmed desorption, suggesting that Fe$_2$O$_3$ possessed negligible oxygen anion defects following a high-temperature calcination. Therefore, Pt atoms disperse across the ferric oxide support where they are anchored as isolated atoms through a strong local surface interaction. Calculations indicate that these reactively-formed Pt atoms coordinate with four surface oxygen atoms in a distorted square geometry with an average Pt–O length of 1.94 Å, intermediate between that in gas PtO$_2$ (CN = 2, 1.70 Å) and bulk PtO$_2$ (CN = 6, 2.04 Å). Bader charge analysis of an isolated Pt

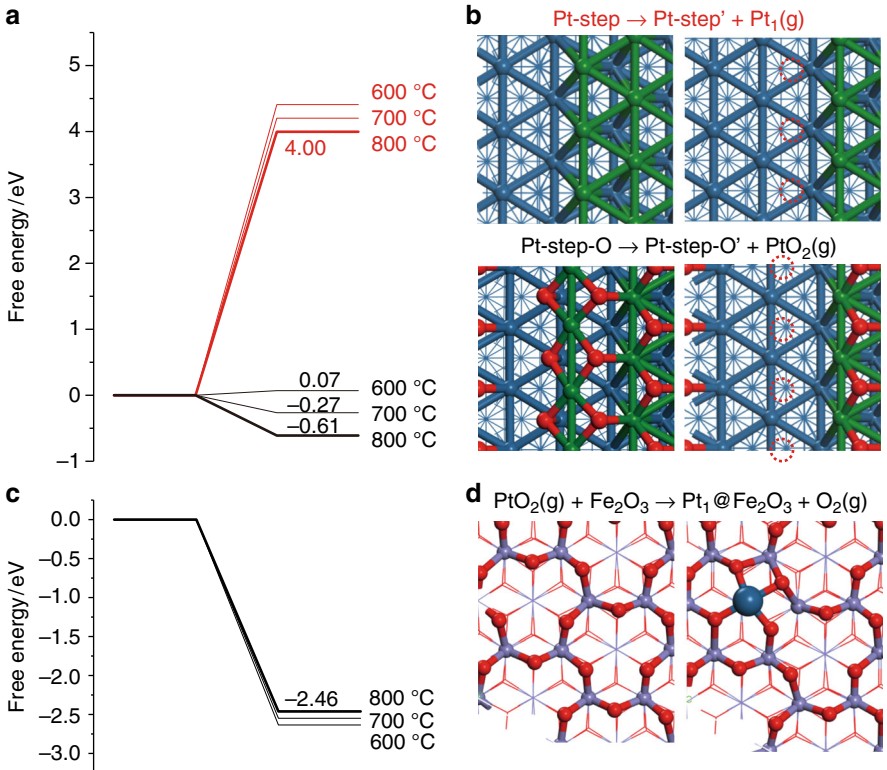

**Fig. 5** Optimized structures and energy profiles for Pt NP dispersion as isolated Pt atoms. **a**, **b** Calculated energies and surface structures for evaporation of a single Pt atom ($Pt_1$) from a Pt(221) step (red value) or evaporation of a $PtO_2$ species from an oxygen pre-covered Pt(221) step (black values). **c**, **d** Calculated energies and surface structures for dissociative capture of $PtO_2$ over $Fe_2O_3$(0001) surface and concomitant formation of a $Pt_1$ atom and evolved $O_2$. Color code: outermost layer Pt (green); second layer Pt (blue); O (red); Fe (purple)

atom on $Fe_2O_3$(0001) shows a charge of $+1.43\ |e|$, approximating to but slightly lower than the charge of Pt in gas $PtO_2$ and bulk $PtO_2$ (Supplementary Table 2), consistent with tetravalent Pt and the XPS analysis (Supplementary Figure 6). The total free energy change for the formation of $Fe_2O_3$ stabilized isolated Pt atoms from Pt NPs is $-3.07$ eV per Pt atom at 800 °C, i.e. such a dispersion is strongly exothermic with low barrier. In comparison, Pt only interacts very weakly (low endothermic free energy) with an alumina surface (Supplementary Figures 11–13, Supplementary Table 3). By comparing the charge density difference of $Pt_1/Fe_2O_3$(0001) and $Pt_1/Al_2O_3$(010), defined as $\Delta\rho = \rho_{Pt+slab} - \rho_{slab} - \rho_{Pt}$, four strong chemical bonds are observed between Pt and adjacent O atoms by covalent ($d$-$p$) orbital interactions at the $Fe_2O_3$ surface. In contrast, only very weak Pt–O interactions occur at the $Al_2O_3$ surface, resulting in Pt–O bond lengths much longer than that on $Fe_2O_3$. Bader analysis indicates that each Pt possesses a $-0.14\ |e|$ charge, i.e. approximately metallic character (Supplementary Table 2); both findings are attributed to the irreducibility of $Al^{3+}$, i.e. strong non-redox Al–O bonding network, which prevents any significant metal-support interaction. DFT calculations are hence in excellent agreement with experimental observations. The finding of the stability of SAC closely related to the reducibility of the support is consistent with previous work as well[55].

**Catalytic performance in methane combustion.** The impact of oxygen-induced restructuring at elevated temperature on catalytic performance is striking (Fig. 6). As the reaction temperature of $1Pt/Fe_2O_3$-NP under an oxygen-rich methane gas feed was raised from 300 to 700 °C, the combustion activity increased such that a modest ~18 % conversion was initially attained. However, in contrast to most catalytic reactions wherein activity subsequently

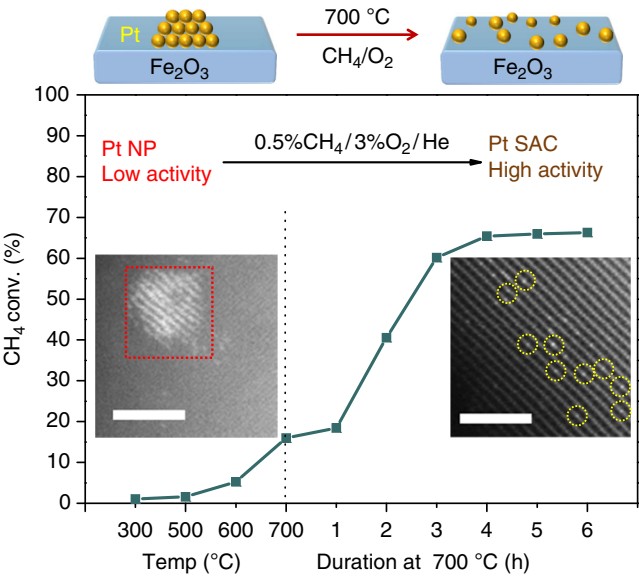

**Fig. 6** Dynamic formation of a Pt SAC during methane oxidation. Light-off curve of $1Pt/Fe_2O_3$-NP for methane oxidation with a feed gas comprising 0.5 vol% $CH_4$/ 3 vol% $O_2$/ 97 vol% He balance at 30 mLmin$^{-1}$. STEM images of catalyst before (left-inset) and after (right-inset) reaction. Scale bars, 2 nm. Red square and yellow circles are used to indicate the Pt NP and Pt atoms, respectively

decreases with time-on-stream, the activity of $1Pt/Fe_2O_3$-NP displayed a monotonic increase over the following 4 h at 700 °C to reach 65 % conversion. Post-reaction catalyst characterization confirmed that this high-temperature catalyst activation was

accompanied by the disappearance of Pt NPs present in the as-prepared material (Fig. 6 left inset), and formation of atomically-dispersed Pt (Fig. 6 right inset, Supplementary Figure 14a–b), demonstrating a causal relationship between in situ genesis of a Pt SAC and dramatically enhanced performance. Note that pre-activation of 1Pt/Fe$_2$O$_3$-NP by calcination at 700 °C for 5 h prior to methane addition resulted in an extremely stable Pt SAC that maintained similar activity for > 16 h on-stream (Supplementary Figure 14c). Since the mass of 1Pt/Fe$_2$O$_3$-NP catalyst (and Pt loading) in this methane light-off experiment is constant, the time-dependent increase in CH$_4$ conversion at 700 °C due to catalyst restructuring is directly proportional to the catalytic activity; single-atom formation induces a four-fold enhancement (18→65 % conversion) in the specific activity. Additional experiments with the same catalyst at 20 % iso-conversion in either its SAC (1Pt/Fe$_2$O$_3$-C700) or nanoparticulate (1Pt/Fe$_2$O$_3$-NP) form confirmed that the specific activity of the former was 4 times greater than that of the latter (2.01 mol$_{CH4}$ h$^{-1}$ g$_{Pt}$$^{-1}$ vs. 0.47 mol$_{CH4}$ h$^{-1}$ g$_{Pt}$$^{-1}$) and 20 times than that reported for Pt/Al$_2$O$_3$[56] (Supplementary Table 4). Turnover frequencies based on the number of surface Pt atoms (dispersion) were similar for SAC and nanoparticulate counterparts (0.1 s$^{-1}$ vs. 0.08 s$^{-1}$), indicating a common active site, and hence the superior specific activity of the single-atom catalyst reflects its improved atom efficiency (every Pt atom directly activates methane).

### Interaction between Pt atoms and iron-modified support.
Fundamental insight into the reaction-induced restructuring of metal NPs over a reducible metal oxide support offers a facile route to synthesize high-loading and thermally stable SACs. Platinum NPs introduced either as pre-formed colloids or e.g. by simple wet impregnation with 1 wt% H$_2$PtCl$_6$ (Supplementary Figure 15a–b) over a low area Fe$_2$O$_3$ support, are readily transformed into isolated Pt atoms by high-temperature calcination (Supplementary Figure 15c–d). Flytzani-Stephanopoulos and co-workers have shown that Na or K ions can stabilize surface hydroxyls, which can in turn stabilize metal (Au and Pt) atoms and/or subnanometer clusters over diverse supports[57–59]. However, our recent work[60] has not identified such a role for alkalis, suggesting that Fe$_2$O$_3$ may have unique properties to some degree. To test our hypothesis, we synthesized a sodium-free material using a (NH$_4$)$_2$CO$_3$ precipitant, denoted as Fe$_2$O$_3$(N). Pt NPs exhibited the same dispersion and single-atom formation after 800 °C calcination over this Fe$_2$O$_3$(N) support (Supplementary Figure 16), confirming that the reducible Fe$_2$O$_3$ support, and not presence of Na$^+$, was responsible for stabilizing Pt single-atoms.

The propensity for stabilizing single Pt atoms is dictated by the magnitude of the metal-support interaction, which itself can be tuned by e.g. doping a reducible (Fe$_2$O$_3$) into a non-reducible (Al$_2$O$_3$) oxide (Supplementary Figure 17a) through co-precipitation. As shown in Fig. 7, functionalization of such an Fe$_2$O$_3$-Al$_2$O$_3$ mixed metal oxide by Pt NPs, and subsequent 800 °C calcination, delivers atomically-dispersed Pt precisely as observed over the pure ferric oxide (Supplementary Figure 17b–c), but in a higher area form (~30 m$^2$ g$^{-1}$).

### Discussion
We reported a facile synthesis of thermally-stable Pt SAC that achieved remarkable catalytic performance towards methane combustion. Isolated Pt atoms with high oxidation state are not stabilized by surface defects but through a strong covalent interaction with iron and oxygen atoms on the surface. Since Fe can be simply incorporated into a variety of oxides with low area and/or defect density, e.g. perovskites and spinels, our approach

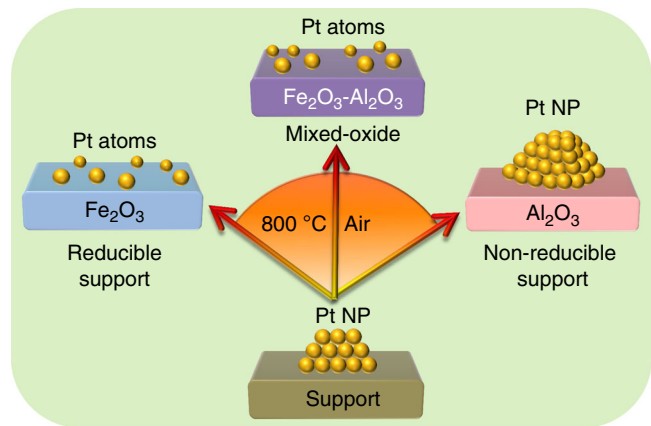

**Fig. 7** Illustration of Pt NP sintering/dispersing on different supports. Metal oxide reducibility dictates the ability of a support to anchor isolated Pt atoms: Fe$_2$O$_3$ favors atomically dispersed Pt, whereas Al$_2$O$_3$ favors nanoparticle sintering. Doping iron oxide into non-reducible support (Fe$_2$O$_3$–Al$_2$O$_3$) can adjust noble metal dispersion

may afford a generic route to fabricate high loading Pt SACs over diverse supports being able to operate under harsh reaction conditions.

### Methods
**Synthesis of FeO$_x$ and Fe$_2$O$_3$.** Ferric nitrate (Fe(NO$_3$)$_3$·9H$_2$O), 98%, Sigma-Aldrich) and sodium carbonate (Na$_2$CO$_3$, 99.5% Sigma-Aldrich) were used as purchased. An aqueous solution of Fe(NO$_3$)$_3$ (1 mol L$^{-1}$) was obtained by dissolving Fe(NO$_3$)$_3$·9H$_2$O (122 g, 0.3 mol) in deionized water (300 mL). FeO$_x$ was subsequently prepared from this solution by precipitation. First, Na$_2$CO$_3$ (11 g) was added in water (100 mL) and stabilized at 50 °C in a water bath. To this, 40 mL Fe (NO$_3$)$_3$ solution was slowly added (1 mL min$^{-1}$) under vigorous stirring at 50 °C for 3 h, and then aged static for a further 2 h. The recovered FeO$_x$ solid was washed with deionized water and dried at 60 °C overnight. Fe$_2$O$_3$ was obtained by calcining FeO$_x$ at 800 °C (ramp rate 3 °C min$^{-1}$) for 5 h under flowing air (100 mL min$^{-1}$).

**Synthesis of Fe$_2$O$_3$(N).** Ferric nitrate (Fe(NO$_3$)$_3$·9H$_2$O), 98%, Sigma-Aldrich) and ammonium carbonate, (NH$_4$)$_2$CO$_3$ Sigma-Aldrich, 99.5%) were used as purchased. First, (NH$_4$)$_2$CO$_3$ (4 g) was added in water (6 mL) and stabilized at 50 °C in a water bath. To this, 20 mL Fe(NO$_3$)$_3$ solution(1 mol L$^{-1}$) was slowly added (1 mL min$^{-1}$) under vigorous stirring at 50 °C for 3 h, and then aged static for a further 2 h. The recovered FeO$_x$ solid was washed with deionized water and dried at 60 °C overnight. Fe$_2$O$_3$(N) was obtained by calcining FeO$_x$ at 800 °C (ramp rate 3 °C min$^{-1}$) for 5 h under flowing air (100 mL min$^{-1}$) to obtain the Na$^+$-free support.

**Synthesis of Fe$_2$O$_3$–Al$_2$O$_3$.** An Fe$_2$O$_3$-Al$_2$O$_3$ mixed oxide was synthesized by co-precipitation. Fe(NO$_3$)$_3$·9H$_2$O (8 g, 0.02 mol) and Al(NO$_3$)$_3$·9H$_2$O (7.5 g, 0.02 mol, 98%, Sigma-Aldrich) were first dissolved in 80 mL deionized water. Separately, (NH$_4$)$_2$CO$_3$ (7.7 g, 0.08 mol, 99%, Sigma-Aldrich) was dissolved in 80 mL deionized water. The two aqueous solutions were combined in 30 mL water at 50 °C under stirring for 3 h, and then aged static for a further 2 h. The resulting solid was recovered by filtration, washed with deionized water, dried at 60 °C overnight, and finally calcined at 800 °C (ramp rate 3 °C min$^{-1}$) for 5 h under flowing air (100 mL min$^{-1}$).

**Synthesis of Pt$_1$/FeO$_x$ and Pt$_1$/FeO$_x$-C800.** Pt$_1$/FeO$_x$ was prepared by co-precipitation of an aqueous mixture of chloroplatinic acid (H$_2$PtCl$_6$·6H$_2$O, 37 mg$_{Pt}$ mL$^{-1}$, 2.6 mL, 99.9%, Sigma-Aldrich) and Fe(NO$_3$)$_3$·9H$_2$O (1 mol L$^{-1}$, 40 mL) with Na$_2$CO$_3$ solution (11 g Na$_2$CO$_3$ in 100 mL H$_2$O) at 50 °C under stirring for 3 h, and ageing static for a further 2 h. The resulting solid was recovered by filtration, washed with deionized water and dried at 60 °C overnight. A portion of the Pt$_1$/FeO$_x$ was then calcined as above at 800 °C and denoted as Pt$_1$/FeO$_x$-C800. The Pt loading determined by ICP was 1.8 wt%.

**Synthesis of $x$Pt/Fe$_2$O$_3$-NP and $x$Pt/Fe$_2$O$_3$-C800.** Pt nanoparticles (NPs) were synthesized according to the literature[61]. Typically, an ethylene glycol solution of NaOH (0.5 mol L$^{-1}$) was added into an ethylene glycol solution of H$_2$PtCl$_6$·6H$_2$O under stirring (3.7 mg$_{Pt}$ mL$^{-1}$). The resulting transparent yellow platinum colloidal solution was heated at 160 °C for 3 h under flowing Ar to form a transparent dark-brown platinum colloidal solution. Separately, Fe$_2$O$_3$ was dispersed in deionized water by sonication, to which the Pt NP colloidal solution was added, and the

resulting solid dried overnight at 100 °C, and then calcined at 500 °C for 5 h under flowing air to remove the glycol stabilizer. A series of Pt NP catalysts were prepared with different Pt loadings, denoted as $x$Pt/$Fe_2O_3$-NP ($x = 0.3$, 1, 2, or 4.5 wt% Pt as determined by ICP). $x$Pt/$Fe_2O_3$-C800 was obtained by treating $x$Pt/$Fe_2O_3$-NP under flowing air (100 mL min$^{-1}$) at 800 °C for 5 h.

**Synthesis of Pt/$Al_2O_3$-NP and Pt/$Al_2O_3$-C800.** Pt/$Al_2O_3$-NP was prepared following the above procedure by dispersing colloidal Pt over the $Al_2O_3$ support at a Pt loading of 0.3 wt% as determined by ICP, and then calcined at 300 °C for 5 h under flowing air. Pt/$Al_2O_3$-C800 was obtained by treating Pt/$Al_2O_3$-NP under flowing air (100 mL min$^{-1}$) at 800 °C for 5 h.

**Synthesis of Pt/$Fe_2O_3$–$Al_2O_3$-NP and Pt/$Fe_2O_3$–$Al_2O_3$-C800.** Pt/$Fe_2O_3$–$Al_2O_3$-NP and Pt/$Fe_2O_3$–$Al_2O_3$-C800 were prepared following the above procedure by dispersing colloidal Pt over the $Fe_2O_3$–$Al_2O_3$ support at a Pt loading of 0.5 wt% as determined by ICP.

**Synthesis of 0.3Pt/$Fe_2O_3$-C600 and 0.3Pt/$Fe_2O_3$-Ar800.** 0.3Pt/$Fe_2O_3$-C600 was obtained by treating 0.3Pt/$Fe_2O_3$-NP under flowing air (100 mL min$^{-1}$) at 600 °C for 5 h. 0.3Pt/$Fe_2O_3$-Ar800 was obtained by treating 0.3Pt/$Fe_2O_3$-NP under flowing Ar (100 mL min$^{-1}$) at 800 °C for 5 h.

**Synthesis of $H_2PtCl_6$/$Fe_2O_3$ and $H_2PtCl_6$/$Fe_2O_3$-C800.** In a typical procedure, 1 g $Fe_2O_3$ was dispersed in 5 mL deionized water by sonication, to which 0.3 mL $H_2PtCl_6$ aqueous solution (37 mg$_{Pt}$ mL$^{-1}$) was added. Water was gradually evaporated by heating the resulting suspension at 80 °C. The resulting solid was calcined at 300 °C for 5 h under flowing air (100 mL min$^{-1}$) and denoted as $H_2PtCl_6$/$Fe_2O_3$. The Pt loading was 1 wt% as determined by ICP. A portion of $H_2PtCl_6$/$Fe_2O_3$ was also heated at 800 °C under flowing air (100 mL min$^{-1}$) for 5 h to generate $H_2PtCl_6$/$Fe_2O_3$-C800.

**Synthesis of $H_2PtCl_6$/$Fe_2O_3$(N) and $H_2PtCl_6$/$Fe_2O_3$(N)-C800.** In a typical procedure, 150 mg $Fe_2O_3$(N) was dispersed in 2 mL deionized water by sonication, to which 0.013 mL $H_2PtCl_6$ aqueous solution (37 mg$_{Pt}$ mL$^{-1}$) was added. Water was gradually evaporated by heating the resulting suspension at 80 °C. The resulting solid was calcined at 300 °C for 5 h under flowing air (100 mL min$^{-1}$) and denoted as $H_2PtCl_6$/$Fe_2O_3$(N). The Pt loading was 0.2 wt% as determined by ICP. A portion of $H_2PtCl_6$/$Fe_2O_3$ was also heated at 800 °C under flowing air (100 mL min$^{-1}$) for 5 h to generate $H_2PtCl_6$/$Fe_2O_3$(N)-C800.

**Synthesis of $H_2PtCl_6$/$Al_2O_3$ and $H_2PtCl_6$/$Al_2O_3$-C800.** α-$Al_2O_3$ (99.5%, Sigma-Aldrich) was used as purchased. $H_2PtCl_6$/$Al_2O_3$ and $H_2PtCl_6$/$Al_2O_3$-C800 were prepared following the above procedure by dispersing $H_2PtCl_6$ over the α-$Al_2O_3$ support. The Pt loading was 1 wt% as determined by ICP.

**Characterization.** Pt loadings were determined by inductively coupled plasma spectrometry-atomic emission spectrometry (ICP-AES) on an IRIS Intrepid II XSP instrument (Thermo Electron Corporation). X-ray diffraction (XRD) patterns were collected on a PW3040/60 X'Pert Pro super (PANalytical) diffractometer, operating at 40 kV and 40 mA using a Cu Kα radiation source ($\lambda = 0.15432$ nm) with a scanning angle ($2\theta$) of 10–80°. In situ diffuse reflectance infrared Fourier transform spectra (DRIFTS) were collected at 25 °C with a Bruker Vertex 70 spectrometer equipped with a mercury cadmium telluride (MCT) detector at a resolution of 4 cm$^{-1}$ over 32 scans. Samples were first heated in situ at 200 °C under flowing He for 0.5 h, then cooled to room temperature prior to recording of background spectra. Subsequently, a 1 vol% CO/He flow was introduced to the sample for 10 min. The environmental cell was then purged with flowing helium to remove gas phase CO, prior to spectral acquisition of chemisorbed CO species. Temperature-programmed desorption (TPD) was conducted on a Micromeritics AutoChem II 2910 automatic catalyst characterization system equipped with a mass spectrometer. First, the $Fe_2O_3$ support was loaded into a U-shape quartz reactor and pretreated at 200 °C in He for 0.5 h to remove adsorbed hydrates and carbonates. After cooling to 60 °C, $NH_3$ gas pulses were injected until a stable signal was obtained, and the sample then heated to 900 °C under He at 10 °C min$^{-1}$. Synchrotron radiation experiments were performed at BL14B2 of SPring-8[62] and BL14W1 of the SSRF light sources. Pt $L_{III}$-edge transmission X-ray absorption spectra were acquired on samples loaded into the $Al_2O_3$ tube, in addition to Pt foil and $PtO_2$ powder references. Spectra were background subtracted, normalized, and fitted using the IFEFFIT software suite[63]. X-ray photoelectron spectra (XPS) were performed on a VG ESCALAB MK2 apparatus with Al Kα radiation (1486.6 eV, 12.5 kV, 250 W) to obtain the binding energies and oxidation states of Pt. Binding energies are referenced to adventitious carbon at 284.8 eV. High-angle annular dark-field scanning transmission electron microscopy (HAADF-STEM) images were obtained on a JEOL JEM-2100F. Aberration-corrected HAADF-STEM images were obtained on a JEOL JEM-ARM200F equipped with a CEOS probe corrector. Samples were dispersed by ultrasonication in ethanol, and the resulting solution dropped on to carbon films supported on copper grids. In situ imaging was performed on a Titan Cubed Themis G2 300 microscope using a DENSsolutions Climate S3 and Gas flow and Heating System, which comprised a heated sample holder and gas delivery manifold. The sample was mounted in a 5 μm gap between two 30 nm thick SiN windows. The oxygen purity used for the in situ calcination experiment was 99.999%. Reported temperatures per the DENSsolutions calibration. Low energy ion scattering (LEIS) spectra were acquired on a Kratos AXIS Supra spectrometer using 1 keV He$^+$ ions generated from a Minibeam 6 Gas Cluster Ion Source, at a scattering angle of 130 °, and 320 eV analyzer pass energy, averaged over a sample area of 500 μm$^2$.

**Catalytic evaluation.** $CH_4$ oxidation was performed in a U-shaped quartz reactor using 20 mg catalyst and a feed gas comprising 0.5 vol% $CH_4$, 3 vol% $O_2$, and 97 vol% He balance at 30 mL min$^{-1}$. The effluent gas composition was analyzed by an on-line Agilent 6890 A gas chromatograph equipped with a TDX-01 column and a thermal conductivity detector. $CH_4$ conversion was calculated based on the difference between inlet and outlet concentrations.

**Turnover frequency (TOF) calculation.** TOFs were calculated from specific activities measured at 700 °C and approximately 20 % $CH_4$ iso-conversion for 17 mg 1Pt/$Fe_2O_3$-NP and 6 mg 1Pt/$Fe_2O_3$-C700 catalysts respectively. A common gas flow rate of 50 mL min$^{-1}$ was used, resulting in a space velocity of 176,000 mL h$^{-1}$ g$_{cat}^{-1}$ for 1Pt/$Fe_2O_3$-NP and 500,000 mL h$^{-1}$ g$_{cat}^{-1}$ for 1Pt/$Fe_2O_3$-C700. The Pt dispersion of 1Pt/$Fe_2O_3$-NP was calculated according to the following relationship between the dispersion (D) and particle radius (r): D / % = 100*5.6/$r_{Pt}$. The average particle radius of 1Pt/$Fe_2O_3$-NP was 15 Å (Fig. 4c), corresponding to a dispersion of 30 %, while that for 1Pt/$Fe_2O_3$-C700 was 100 %. TOFs were calculated by normalizing specific activities to the product of the dispersion and total Pt$_{mols}$ (identical for both catalysts).

**DFT parameters.** All DFT + U calculations were performed using spin-polarized density functional theory (DFT) with the generalized gradient approximation (GGA) and Perdew-Burke-Ernzerhof (PBE) exchange-correlation functional as implemented in VASP5.3.5[64–66]. The U-J value of 3.0 eV was used to describe the strong correlation of the localized Fe 3d states[3,67,68]. Valence states of all atoms were expanded in a plane wave basis set with a cutoff energy of 400 eV, and a Monkhorst-Pack mesh of $3 \times 3 \times 1$ k points was used for Brillouin Zone integration. Atomic positions were optimized by the conjugate gradient algorithm until forces were < 0.02 eV/Å. Free energy changes were calculated from zero-point energies, enthalpies, and entropies of gas Pt single atom and $PtO_2$ species corrected at 600, 700 and 800 °C, respectively as summarized in Supplementary Table 3.

**Computational models.** A Pt(221)-p($3 \times 1$) surface slab was used to model the step and edge of Pt NPs (coordination number = 7 for atoms at edges). We applied periodic models of surface slabs with five atomic layers, with the bottom two layers frozen and the remaining layers allowed to relax. The (0001) surfaces of α-$Fe_2O_3$-p ($2 \times 2$) were also represented by a periodic slab model. Antiferromagnetic properties of α-$Fe_2O_3$ were represented by a ( + − − + ) magnetic configuration, which was previously proven to be most energetically favorable for α-$Fe_2O_3$[3]. We chose slabs containing 9 layers of Fe atoms and 5 atomic layers of O to model the O-terminated surface. The bottommost Fe–O layers were frozen during geometry optimization. A $\theta$-$Al_2O_3$(010)-p($4 \times 2$) surface slab was used to model the alumina substrate, consisting of six O–Al layers, wherein the bottom two O−Al layers were frozen while the remaining layers were allowed to relax[69,70]. The IDIPOL tag was set to 3 to switch on dipole corrections to the total energy along the z-direction. All supercell slabs were periodically repeated with a 15 Å vacuum layer between surfaces in the direction of the surface normal.

**Theoretical maximum loading of dispersed Pt atoms over $Fe_2O_3$ support.** The BET surface area of $Fe_2O_3$ was 5–10 m$^2$ g$^{-1}$, hence 1 g of $Fe_2O_3$ provides ~5–10 m$^2$ of surface (S). Our DFT calculation model indicates that the maximum density of atomically dispersed Pt (D) is 4.5 atom nm$^{-2}$. The total number of isolated Pt atoms (N) that could be achieved for 1 g of Pt/$Fe_2O_3$ is therefore predicted to be $N = D \times S$. Since the mass of Pt (m) equals N/$N_A$ × M, where $N_A$ is Avogadro's constant ($6.02 \times 10^{23}$ mol$^{-1}$), and M is the molar mass of Pt (195 g mol$^{-1}$), the theoretical maximum loading of isolated Pt atoms that could be dispersed over 1 g of $Fe_2O_3$ is m(g)/1(g) × 100 % = ($D \times S$ / $N_A$) × M × 100% = 1.5 wt%.

## Data availability

The data that support the plots within this paper and other findings of this study are available from the corresponding authors upon reasonable request.

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

## Acknowledgements

This paper is dedicated to the 70th anniversary of the Dalian Institute of Chemical Physics, Chinese Academy of Sciences. This work was supported by National Natural Science Foundation of China (21606222, 21573232, 21776270, 21590792, 91645203, 51761165012), National Key Projects for Fundamental Research and Development of China (2016YFA0202801, 2017YFA0700104), Strategic Priority Research Program of the Chinese Academy of Sciences (XDB17020100), and DNL Cooperation Fund, CAS (DNL180403). The synchrotron radiation experiment was performed at the BL14B2 of SPring-8 with the approval of Japan Synchrotron Radiation Research Institute (Proposal No. 2011B1974), Japan, and the BL14W1 at the Shanghai Synchrotron Radiation Facility, Shanghai Institute of Applied Physics, China. We thank Kratos Analytical Ltd, Manchester for LEIS measurements. Financial grant from the China Postdoctoral Science Foundation (2017M621170). DICP Outstanding Postdoctoral Foundation (2017YB02), and dedicated funds for methanol conversion from DICP. J. Luo thanks National Program for Thousand Young Talents of China, Tianjin Municipal Science and Technology Commission (15JCYBJC52600). The authors thank Kai Wang, Mengke Ge and Xingang Hou for their assistance with in situ STEM measurements. The computational work was performed using supercomputers at Tsinghua National Laboratory for Information Science and Technology and Lvliang Tianhe-2 Supercomputing Center.

## Author contributions

R.L. synthesized the catalyst and performed most of the reactions. S.M. performed the electron-microscopy characterization. W.X. and J.Luo performed the in situ electron microscopy. J-C.L. and J.Li did the theoretical calculations and analysis. Y-T.C., X.L. and Lei Li carried out the EXAFS experiment. Lin Li carried out the CO-DRIFT experiment. Y.C. and J.Lin provided reagent. T.L. and F.C. performed some experiments. A.F.L. provided the LEIS data and edited the manuscript. R.L. and B.Q. wrote the manuscript. A-Q.W. and X.W. revised the paper. B.Q. and T.Z. designed the study and supervised the project.

## Additional information

**Competing interests:** The authors declare no competing interests.

