## [Peer Review File · Nature Communications]

Reviewers' Comments:

Reviewer #1:

Remarks to the Author:
review attached

the paper needs significant revision

Reviewer #2:

Remarks to the Author:

The principal claim of the paper is that single atom Pt has been formed on uniform non-defect sites on FeOx. This claim is based on TEM images which show the atomic dispersion and on DFT calculations which show Pt atoms coordinated to 4 O atoms. There is no experimental evidence of this coordination as could be determined [with some error] by XAFS spectroscopy, there is no evidence that there are not some Pt atoms on other sites [minority sites] and there is no evidence of uniformity of sites that would be shown by narrow CO infrared bands as in the work by Christopher in J. Am. Chem. Soc. 2017 [the bands shown are not at all narrow like Christopher's]. The broad bands are evidence of nonuniform structures. The DFT calculations could have been done for competing structure models. The method of preparation of the single atom Pt is close to the method described to make ceria supported Pt [as cited]. The methane conversion catalysis data are not presented in a thorough fundamental way with TOF values. Statement on pg 2 that SAC stability is intrinsically linked to the density and stability of defects is not justified, ignores lots of literature and is not supported by literature generally.

Reviewer #3:

Remarks to the Author:

In this paper, Qiao et al. have shown the redispersion of Pt species into single Pt atoms on FeOx and FeOx-Al₂O₃ mixed oxide support. The results shown in this manuscript are meaningful to understand the dynamic behavior of supported metal catalysts under redox and reaction conditions. The following comments should be considered before being accepted for publication in Nature Communications.

1) EXAFS results should be provided to show the full dispersion of Pt nanoparticles into Pt single atoms, including the samples shown in Figure 1 and Figure S4).

2) It is found that, Pt nanoparticles cannot be redispersed on Al₂O₃ after high-temperature calcination in air. If comparing the initial H₂PtCl₆/Al₂O₃ (shown in Figure S7) and H₂PtCl₆/Fe₂O₃ (shown in Figure S13), the sizes of Pt particles are different in those two cases. The initial size of Pt particles in the H₂PtCl₆/Fe₂O₃ sample is smaller. It can be expected that, the redispersion dynamic of Pt nanoparticles into Pt single atoms are related with their particle size, which has also been reported in a recent paper (Nature Communications 9 (1), 574.). Therefore, to have a fair comparison, I would like to suggest to author to load the Pt-PVP nanoparticles on Al₂O₃ (as done with Fe₂O₃, in Figure S3) and then treat the sample by calcination in air at 800 oC to see if those Pt nanoparticles can be re-dispersed on Al₂O₃.

3) In the title of this manuscript, the authors claim that, the redispersion of Pt nanoparticles into single atoms occur on "non-defect" support. I agree that, compared to CeO₂ or TiO₂, Fe₂O₃ may have less defects, such as oxygen vacancies. However, there are still other types of defects on the surface

of Fe₂O₃ nanoparticles, due to the roughness of the surface (see: Journal of Catalysis 324 (2015) 127–132. Journal of Electron Microscopy, 58 (2009) 199–212.). Those unsaturated sites can be anchoring sites for Pt atoms. According to the data presented in this manuscript, the positions of Pt atoms on Fe₂O₃ cannot be determined. Therefore, I will suggest the authors not to use "non-defect" to describe the support and the corresponding redispersion behaviour unless they can provide more proof or more discussion on that point.

4) The in situ TEM experiments have been carried out to investigate the redispersion behaviour of Pt nanoparticles on Fe₂O₃, as presented in Figure 2. Please, make a careful look into the images, it seems that some of the nanoparticles (especially in the middle area) slightly grow. Does it mean the Ostwald ripening mechanism also occurs? The size distributions of the Pt nanoparticles should be provided for the fresh sample and the sample after 20 min at 800 °C to see the structural evolution.

5) According to the experimental details for preparation of FeO_x and Fe₂O₃, it seems that, there is a residual amount of sodium in the final solid carrier, as reported in the recent publication from their group (Chem. Sci., 2017,8, 5126-5131). Does it come from the Al₂O₃ support?. Does sodium play any role in the stabilization of single Pt atoms during the redispersion process? Control experiments using Na-free Fe₂O₃ as the support are recommended.

Response to Reviewer 1

This paper describes a predominantly experimental study of the generation and properties of single atom Pt dispersed on Fe₂O₃. The authors provide convincing evidence for the observed phenomena and use DFT studies to help understand some of the energetics behind what they observe. The experimental work is of high quality and they do seem to demonstrate the presence of site isolated metal atoms. However, there are some real issues to address before acceptance.

1. Line 37. There is no such thing as ‘atom thrifting’. Rewrite.

Response: While we agree that “atom thrifting” is a confusing term, the concept of “precious metal thrifting” is well-established in the catalysis and sustainability sectors, see e.g. Mooiman, M. B., Sole, K. C. and Dinham, N. (2016). *The Precious Metals Industry in Metal Sustainability*, R. M. Izatt (Ed.)¹. Thrifting features extensively in industry reports by Johnson Matthey spanning decades (*Platinum Metals Rev.* **1985**, 29, 2²; *Johnson Matthey Technol. Rev.* **2018**, 62, 429³), and the academic literature (*Curr. Opin. Electrochem.* **2018**, 9, 271⁴). We therefore prefer to keep this increasingly common terminology. A reference (ref. 6) to precious metal thrifting is now included in the introduction.

Action: Manuscript amended

2. Add the following recent reference to the Introduction: “Atomically dispersed supported metal catalysts: perspectives and suggestions for future research,” B. C. Gates, M. Flytzani-Stephanopoulos, D. A. Dixon, and A Katz, *Catal. Sci. Technol.*, 2017, 7, 4259-4275.

Response: Thank you for your suggestion. We have cited this paper as reference 29 in our revision and now include an extensive list of recent reviews on the role of support defects in stabilizing single-atom catalysts as a new Supplementary Appendix III.

Action: Manuscript+ESI amended

3. Line 143. Start line with ‘The’

Response: Thank you for the suggestion, “The” has been added in the revision.

Action: Manuscript amended

4. For the computational work, explain how they get free energies using VASP as this requires vibrational partition functions for solid and gas phase species. This is not trivial to do. Please write out reactions for the DFT section as I am not sure what they are actually calculating in terms of how PtO₂ is formed and released as well as the re-addition/distribution of PtO₂ on the surface. It is hard to figure out what they have actually calculated.

Response: The reviewer raises an important aspect regarding our free energy calculations, and the relevant details are now included in the revised supplementary information (new Supplementary Table S3), and below.

The chemical equation of evaporation of one row of Pt or one row of PtO₂, i.e. three Pt or three PtO₂ in each slab model, at step of Pt(221) can be written as:

The chemical equation for adding PtO₂ to Fe₂O₃(0001) or θ-Al₂O₃(010) surfaces is:

Thus, the energy changes at 0 K (neglecting zero point energy (ZPE)) are:

$$\Delta E(1) = (E_{\text{Pt}(221 \text{ slab, with one row Pt evaporated})} + 3E_{\text{Pt}(\text{gas})} - E_{\text{Pt}(221 \text{ slab})})/3 = 6.00 \text{ eV}$$

$$\Delta E(2) = (E_{\text{Pt}(\text{oxygen covered 221 slab, with one row PtO}_2 \text{ evaporated})} + 3E_{\text{PtO}_2(\text{gas})} - E_{\text{Pt}(\text{oxygen covered 221 slab})})/3 = 2.38 \text{ eV}$$

$$\Delta E(3) = E_{\text{Pt}_1@\text{Fe}_2\text{O}_3} + E_{\text{O}_2(\text{g})} - E_{\text{Fe}_2\text{O}_3} + E_{\text{PtO}_2(\text{g})} = -3.16 \text{ eV}$$

$$\Delta E(4) = E_{\text{Pt}_1@\text{Al}_2\text{O}_3} + E_{\text{O}_2(\text{g})} - E_{\text{Al}_2\text{O}_3} + E_{\text{PtO}_2(\text{g})} = -0.07 \text{ eV}$$

The standard Gibbs free energies of $G_{\text{O}_2(\text{g})}$, $G_{\text{Pt}(\text{g})}$, and $G_{\text{PtO}_2(\text{g})}$ were calculated using the following equations, taking into account the individual translational E_t and S_t , vibrational E_v and S_v , rotational E_r and S_r , and ZPE contributions:

$$G = H - TS = U + k_b T - TS$$

$$S = S_t + S_v + S_r$$

$$U = E_{DFT} + ZPE + E_t + E_v + E_r$$

where E_{DFT} are the electronic energies from DFT calculations. S_t , S_v , S_r , E_t , E_v , and E_r are obtained by including partition functions, Q , according to:

$$U = k_b T^2 \left(\frac{\partial \ln Q}{\partial T} \right)_{N,V}$$

$$S = k_b \ln Q + k_b T \left(\frac{\partial \ln Q}{\partial T} \right)_{N,V}$$

$$\ln Q = N \left[\ln \left(\frac{q_{trans}}{N} \right) + 1 \right] + N \ln q_{rot} + N \ln q_{vib} + N \ln q_{ele}$$

For slab models, the entropy and enthalpy corrections to free energies are neglected in this work. The resulting corrections to the ZPE, H , S , and G at various temperatures are given below:

		ZPE (eV)	$H_{0 \rightarrow T}$ (eV)	S (eV/K)	$G_{0 \rightarrow T}$ (eV)
Pt(gas)	873 K	0.00	0.19	0.00204	-1.59
	973 K	0.00	0.21	0.00206	-1.80
	1073 K	0.00	0.23	0.00208	-2.01
PtO ₂ (gas)	873 K	0.14	0.60	0.00333	-2.31
	973 K	0.14	0.66	0.00340	-2.65
	1073 K	0.14	0.72	0.00346	-2.99
O ₂ (gas)	873 K	0.10	0.38	0.00248	-1.79
	973 K	0.10	0.41	0.00252	-2.04
	1073 K	0.10	0.45	0.00256	-2.29

ΔG for the preceding reactions can be then estimated by adding the above corrections to ΔE , resulting in the following figures:

Action: Manuscript+ESI amended

- The authors should remember that catalysis is about rate enhancement, not just percent conversion. How much is the rate enhanced? Are they just making CO₂ or are there value added products? Combustion of CH₄ to CO₂ at 700 C is not an exciting reaction. What are they trying to show here? Did they detect the CO₂ and H₂O as products of the reaction to match the conversion of the CH₄?

Response: The reviewer is entirely correct that conversion is rarely a useful measure of catalytic activity *when comparing different catalysts* since a change in conversion may simply reflect an

accompanying change in reactant:catalyst (and/or active site) ratio. However, in the present work, we demonstrate the impact of Pt nanoparticle dispersion and associated single-atom formation for the *same catalyst* in an operando model (Figure 5). Since the mass of **1Pt/Fe₂O₃-NP** catalyst (and Pt loading) in our methane light-off experiment is constant, the increase in CH₄ conversion at 700 °C with time due to catalyst restructuring is directly proportional to the catalytic activity; single-atom formation induces a four-fold enhancement (18→65 % conversion) in the specific activity. Additional experiments with the same catalyst at 20 % iso-conversion in either its single-atom (1Pt/Fe₂O₃-C700) or nanoparticulate (1Pt/Fe₂O₃-NP) form confirmed that the specific activity of the former was 4 times greater than that of the latter (2.01 mol_{CH₄}·h⁻¹·g_{Pt}⁻¹ versus 0.47 mol_{CH₄}·h⁻¹·g_{Pt}⁻¹). These results appear in a new Supplementary Table S4. Turnover frequencies based on the number of surface Pt atoms (dispersion) were similar for single-atoms and nanoparticles (0.1 s⁻¹ vs 0.08 s⁻¹), indicating a common active site, and hence the superior specific activity of the single-atom catalyst reflects its improved atom efficiency (every Pt atom directly activates methane).⁵

We respectfully disagree that methane combustion to CO₂ is not an ‘interesting’ reaction; natural gas power plants accounted for ~25 % of global energy production in 2015⁶ and their contribution is expected to rise over coming decades. Methane combustion is thus one of the primary means of human energy production, and least polluting of fossil fuels. Although the reaction temperature of 700 °C required for in situ genesis of our single-atom catalyst (SAC) is high compared to those employed in commercial catalytic methane combustion, the specific activity of our Pt₁/Fe₂O₃ catalyst compares very favorably with the literature,⁷ and is ~20 times that of conventional Pt/Al₂O₃ (0.093 mol_{CH₄}·h⁻¹ g_{Pt}⁻¹ at 700 °C) as seen in the new Supplementary Table S4. GC analysis revealed CO₂ as the sole reaction product (water could not be measured with this column).

Action: Manuscript+ESI amended

6. After the catalytic reaction, was the catalytic material recovered and analyzed to see if they ended up with exactly what they started with? Is the catalyst unchanged at the end of 6 hours

at 700 C? These issues need to be addressed. What other catalyst are they comparing to in terms of performance?

Response: Post-reaction characterization was performed on the Fe₂O₃ supported Pt colloidal NP catalyst (1Pt/Fe₂O₃-NP) from Figure 5, which was subjected to 6 h methane combustion at 700 °C. The resulting STEM confirmed that the initial NPs were completely dispersed into single atoms and remained atomically-dispersed for extended periods at high temperature (Figure 5, and Supplementary Figure S14). We also compared our catalyst with a previously reported Pt/Al₂O₃ catalyst,⁷ which reveals our Pt₁/Fe₂O₃ SAC is 20 times more active (Supplementary Table S4). These aspects are now described in the manuscript and supplementary information.

Action: Manuscript+ESI amended

7. They state that Pt does not bind to Al₂O₃ very well. Did they do calculations to compare the Fe₂O₃ and Al₂O₃ to see what the energetic differences are? What is the driving energy for the stabilization of Pt on Fe₂O₃? They do not provide this nor do they provide density of states, for example, for the Fe₂O₃ and Al₂O₃ to contrast them to show what is happening. The calculations may support the results but they did not do any calculations to show why Al₂O₃ does not work well. This is needed to backup their assertions about why the two metal oxides are different. What if the difference is due to thermodynamic differences in metal-oxygen bond energies? They need to do the same calculations for the Al₂O₃ as done for the Fe₂O₃. What form of Al₂O₃ are they describing?

Response: The reviewer raises an interesting question regarding the differences between Al₂O₃ and Fe₂O₃. We indeed compared the energetics of these systems, as shown in Supplementary Figures S11 and Supplementary Table S2, and now include additional data (Figures S12-13) and associated analysis and discussion on the stabilization of Pt over Fe₂O₃ versus Al₂O₃ in the supplementary information as outlined below (and in the manuscript).

The driving energy for the stabilization of Pt on Fe₂O₃ arises from Pt-O bond formation. As shown in Supplementary Figure 12, we calculate the charge density difference of Pt₁/Fe₂O₃(0001) and Pt₁/Al₂O₃(010), defined as $\Delta\rho = \rho_{\text{Pt+slab}} - \rho_{\text{slab}} - \rho_{\text{Pt}}$. For Pt₁/Fe₂O₃(0001), there are four strong chemical bonds between Pt and adjacent O resulting from d-p orbital interactions, which

result in an oxidized Pt with +1.43 |e| from Bader charge analysis. Here +1.43 |e| is slightly lower than the formal charge of Pt in gas PtO₂ and bulk PtO₂ (Supplementary Table 1). However, the Pt-O bond lengths on the Fe₂O₃ surface are around 1.94 Å, even shorter than in crystalline PtO₂. In contrast, Pt-O bonds on the Al₂O₃ surface are much longer at 2.35 Å, evidencing a far weaker interaction with surface O atoms.

Supplementary Figure 12. Calculated charge density differences for Pt adatoms on the (A) Fe₂O₃(0001) and (B) Al₂O₃(010) surface. Yellow and blue areas represent charge increase and reduction, respectively. The cut-off of the density-difference isosurfaces equals 0.005 electrons/Å³. (C, D) Two dimensional representation of charge difference at the Pt horizontal face.

Supplementary Table S2. Bader charge analysis of Pt atoms, and Pt-O bond lengths for gas and condensed phase PtO₂, Pt/Fe₂O₃, and Pt/Al₂O₃.

	PtO ₂ (gas)	PtO ₂ (bulk)	Pt/Fe ₂ O ₃	Pt/Al ₂ O ₃
Bader charge of Pt / e⁻	+1.61	+1.73	+1.43	-0.14
Pt-O bond length / Å	1.70	2.04	1.90, 1.94, 1.95, 1.95	2.35, 2.36

Comparing the projected electronic density of states (PDOS) for Fe₂O₃(0001) and Al₂O₃(010) as shown in Supplementary Figure 13, the s and p orbitals of O²⁻ at the Al₂O₃(010) surface are fully occupied, and the band gap of Al³⁺ is too large to accept electrons from Pt, i.e. the strong Al-O bonding network prevents any significant metal-support interaction with Pt adatoms. This

contrasts with O^{n-} at the $Fe_2O_3(0001)$ surface, whose orbitals are not fully occupied, and reducibility of Fe^{m+} ions, which remain to be able to accept electrons from Pt adatoms; factors favoring a strong metal-support interaction.

Supplementary Figure S13. Projected electronic density of states (PDOS) of O, Al, Fe, and Pt on (A) $Al_2O_3(010)$, (B) $Fe_2O_3(0001)$, (C) $Pt_1/Al_2O_3(010)$, and (D) $Pt_1/Fe_2O_3(0001)$ surfaces.

Action: Manuscript+ESI amended

8. In my opinion, figure S5 should be in the text as it not supplementary data but is a model.

Response: We thank the reviewer for this suggestion and have now moved the figure into the text as Scheme 1 in the revised manuscript.

Action: Manuscript+ESI amended

9. Lines 113-125 in the SI should be in the text, perhaps in the experimental methods.

Response: We thank the reviewer for this suggestion and this text has now been moved to the Methods section as “Theoretical maximum loading of dispersed Pt atoms over Fe₂O₃ support”.

Action: Manuscript+ESI amended

10. Figure S10. Please use a white background instead of black as this figure is hard to see. As this is SI, expand the figures so they are not so small. For the calculations, provide either references or benchmarks as to their quality.

Response: The reviewer makes an excellent suggestion. All panels in Figure S10 now feature white backgrounds.

Supplementary Figure S10. Computational model of Fe₂O₃(0001), Pt(221) step, and θ -Al₂O₃(010) surfaces.

Calculated models and parameters for θ -Al₂O₃(010) are derived from (*J. Am. Chem. Soc.*, 2013, 135, 12634-12645) and (*J. Phys. Chem. C* 2012, 116, 5628). Calculated model and parameters

for α -Fe₂O₃(0001) are derived from (*Nat. Chem.*, 2011, 3, 634-641). These references are now cited in the appropriate Methods section of the manuscript (ref 60, 61, and 3 respectively).

Action: Manuscript+ESI amended

Response to Reviewer 2

The principal claim of the paper is that single atom Pt has been formed on uniform non-defect sites on FeOx. This claim is based on TEM images which show the atomic dispersion and on DFT calculations which show Pt atoms coordinated to 4 O atoms. There is no experimental evidence of this coordination as could be determined [with some error] by XAFS spectroscopy, there is no evidence that there are not some Pt atoms on other sites [minority sites] and there is no evidence of uniformity of sites that would be shown by narrow CO infrared bands as in the work by Christopher in J. Am. Chem. Soc. 2017 [the bands shown are not at all narrow like Christopher's]. The broad bands are evidence of nonuniform structures. The DFT calculations could have been done for competing structure models. The method of preparation of the single atom Pt is close to the method described to make ceria supported Pt [as cited]. The methane conversion catalysis data are not presented in a thorough fundamental way with TOF values. Statement on pg 2 that SAC stability is intrinsically linked to the density and stability of defects is not justified, ignores lots of literature and is not supported by literature generally.

Response: We thank the reviewer for the comments and suggestions, although we respectfully disagree with some of these as discussed below.

Experimental evidence for Pt adatoms coordinated to 4 surface oxygens in our SAC is in fact strong and clearly evidenced by EXAFS fitting (Supplementary Figure S5B and Supplementary Table S1), with bond lengths in good agreement with our DFT model. The corresponding Debye-Waller factor for this Pt-O₄ environment is only 0.0004, i.e. a very small error. We have added this new data and analysis into the revised manuscript.

The reviewer is undoubtedly correct that every isolated Pt atom does not reside in an identical environment on the Fe₂O₃ support, and we did not claim that this was the case. Indeed, since our Fe₂O₃ was synthesized by a scalable wet chemical (co-precipitation) method, by which it is impossible to obtain monodispersed oxide nanoparticles with identical morphologies and defect densities, it would be remarkable if every Pt atom occupied an identical local surface environment. Nevertheless, EXAFS is consistent with the majority of isolated Pt atoms coordinating to four oxygen nearest neighbours. That our CO IR bands are broader than those of Christopher and co-workers study⁸ is also unsurprising, since the essence of their elegant work was to use ultra-low Pt loadings (0.05 wt%) to immobilize only one Pt atom per 5 nm

monodispersed anatase nanocrystal. Such low loadings favor the population of only the most reactive titania surface sites, and hence excellent homogeneous local environments. A brief comment on this aspect is now included in the manuscript. Other supported single-atom catalyst studies exhibit much broader CO IR bands (see e.g. Pt/CeO₂ in Fig. 2 of *Science* **2017**, 358, 1419⁹, and Fig. 1e of *ACS Catal.* **2017**, 7, 887¹⁰). Any small variations in the precise local coordination environment of isolated Pt atoms in our high loading Pt/Fe₂O₃ SAC are therefore of far less scientific significance than the extremely high specific activities offered by our new synthetic protocol.

We agree that methane conversion data alone are not sufficient to describe the intrinsic reactivity of our SAC, and now report and briefly describe the corresponding TOFs in the manuscript and a new Supplementary Table S4 our single-atom catalyst has a specific activity about 4 times greater than that of the NP catalyst (2.01 mol_{CH₄}·h⁻¹ g_{Pt}⁻¹ vs 0.47 mol_{CH₄}·h⁻¹·g_{Pt}⁻¹) but a similar TOF (0.1 s⁻¹ vs 0.08 s⁻¹).

We respectfully disagree with the reviewer's statement that there is little evidence SAC stability is intrinsically linked to the density and stability of defects. Numerous experimental and DFT studies show and/or imply the link between SAC stability and defect density/stability as now highlighted in new Supplementary Appendices I and II:

Supplementary Appendix I: Experimental studies of single atoms located at support defects.

Entry	sample	location	ref
1	Pd/MgO	O vacancy	11
2	Pt/Al ₂ O ₃	Al ³⁺ _{penta}	12
3	Pd/Al ₂ O ₃	defect	13
4	Fe/SiO ₂	Si vacancy	14
5	Pt/C	defect	15
6	Pd/C	defect	16
7	Pt/C	vacancy	17
8	Fe,Co,Ni/C	vacancy	18
9	Pd/C	vacancy	19
10	Co-Pt/C	defect	20
11	Pd/mpg-C ₃ N ₄	“six-fold Cavities”	21
12	Ir,Au,Pd,Ag,Pt/C	defect	22
13	Pd/graphene	vacancies	23
14	Pt/graphene	defects	24
15	Pt,Co, In/graphene	vacancies	25
16	Pt/MoS ₂	S vacancy	26

17	Co/MoS ₂	defect	27
18	Pt/CeO ₂	defects	28
19	Au/CeO ₂	O defect	29
20	Au/CeO ₂	O vacancy	30
21	Au/CeO ₂	Ce vacancy	31
22	Pt/CeO ₂	step edges	32
23	Pt/CeO ₂	Nano pocket	33
24	Pt/CeO ₂	O vacancy	34
25	Pt/FeO _x	O vacancy	35
26	Co/Fe ₃ O ₄ (001)	octahedral vacancies	36
27	Ni/Fe ₃ O ₄ (001)	cation vacancies	37
28	Cu, Ag/Fe ₃ O ₄ (001)	“narrow” site	38
29	Au/Fe ₃ O ₄ (001)	“narrow” hollow site	39
30	Ni, Co, Mn, Ti, and Zr/Fe ₃ O ₄	cation vacancy	40
31	Rh/CoO	O vacancy	41
32	Pt/Ni(OH) _x	Ni ²⁺ vacancy	42
33	Pt, Au/ZnO	Zn vacancy	43
34	Rh/ZnO	vacancy	44
35	Au/TiO ₂	O vacancy	45
36	Pt/TiO ₂	Defect	8
37	Pt/TiO ₂	O vacancy	46
38	Pd/TiO ₂	defects	47
39	Pt/WO _x	O vacancy	48

Supplementary Appendix II: DFT studies of single atoms located at support defects.

Entry	sample	location	ref
1	Pt/NB	defects	49
2	Pd/NB	B vacancy	50
3	Fe/MoS ₂	S vacancy	51
4	Rh/CoO	O vacancy	52
5	Co/graphene	vacancy	53
6	Au/graphene	defect	54
7	M/FeO _x (M = Au, Rh, Pd, Co, Cu, Ru and Ti)	O vacancy	55
8	Au/CeO ₂	O vacancy	56
9	Au/CeO ₂	O defect	57

Deliberately creating defects in solid supports has evolved as an effective method to deposit isolated metal atoms, and the relevant content summarized in review articles in Appendix III. We apologize for not citing all of these elegant works in the manuscript due to reference limitations.

Supplementary Appendix III: Reviews citing the localization of single atoms at support defects.

Entry	title	ref
1	Preparation, characterization and catalytic performance of single-atom catalysts.	58
2	Increasing the range of non-noble-metal single-atom catalysts.	59
3	Two-dimensional materials confining single atoms for catalysis.	60
4	Atomically Dispersed Supported Metal Catalysts.	61
5	Single-Atom Catalysts: A New Frontier in Heterogeneous Catalysis.	62
6	Catalysis by Supported Single Metal Atoms.	63
7	The Power of Single-Atom Catalysis.	64
8	Atomically dispersed supported metal catalysts: perspectives and suggestions for future research.	65
9	Single-Atom Electrocatalysts.	66
10	Metal Catalysts for Heterogeneous Catalysis: From Single Atoms to Nanoclusters and Nanoparticles.	67
11	Strategies for Stabilizing Atomically Dispersed Metal Catalysts.	68
12	Single-Atom Catalysts: Emerging Multifunctional Materials in Heterogeneous Catalysis	69
13	Heterogeneous single-atom catalysis.	70
14	Unravelling single atom catalysis: The surface science approach.	71

Action: Manuscript+ESI amended

Response to Reviewer 3

In this paper, Qiao et al. have shown the redispersion of Pt species into single Pt atoms on FeOx and FeOx-Al₂O₃ mixed oxide support. The results shown in this manuscript are meaningful to understand the dynamic behavior of supported metal catalysts under redox and reaction conditions. The following comments should be considered before being accepted for publication in Nature Communications.

1) EXAFS results should be provided to show the full dispersion of Pt nanoparticles into Pt single atoms, including the samples shown in Figure 1 and Figure S4).

Response: We thank the reviewer for the suggestion and now include EXAFS data for both samples in Figure 1E and a new Figure 2. EXAFS fits and fitted parameters for the 0.3Pt/Fe₂O₃-C800 sample appear in a new Figure S5 and Supplementary Table S1. All these EXAFS data evidence the complete dispersion nanoparticles as isolated Pt atoms, with only Pt-O nearest neighbor scatters (no Pt-Pt contributions).

Action: Manuscript+ESI amended

2) It is found that, Pt nanoparticles cannot be redispersed on Al₂O₃ after high-temperature calcination in air. If comparing the initial H₂PtCl₆/Al₂O₃ (shown in Figure S7) and H₂PtCl₆/Fe₂O₃ (shown in Figure S13), the sizes of Pt particles are different in those two cases. The initial size of Pt particles in the H₂PtCl₆/Fe₂O₃ sample is smaller. It can be expected that, the redispersion dynamic of Pt nanoparticles into Pt single atoms are related with their particle size, which has also been reported in a recent paper (Nature Communications 9 (1), 574.). Therefore, to have a fair comparison, I would like to suggest to author to load the Pt-PVP nanoparticles on Al₂O₃ (as done with Fe₂O₃, in Figure S3) and then treat the sample by calcination in air at 800 °C to see if those Pt nanoparticles can be re-dispersed on Al₂O₃.

Response: The reviewer makes an excellent suggestion. We have therefore loaded 0.3 wt% of colloidal Pt NPs on Al₂O₃ and calcined these at 800 °C in flowing air precisely as for Fe₂O₃

(synthetic procedure in the Methods- Pt/Al₂O₃-NP and Pt/Al₂O₃-C800). The resulting TEM images of this Pt/Al₂O₃-NP material before and after calcination now appear in Figure S7 E-H. The initial Pt particle size was about 2-3 nm (the same as our Pt/Fe₂O₃-NP catalyst). Following calcination, severe nanoparticle aggregation was observed resulting in Pt agglomerates >10 nm. This experiment confirms that Pt nanoparticles cannot be dispersed on Al₂O₃ by high temperature calcination, irrespective of their initial particle size. We note that dispersion of <1 nm Pt NPs was recently reported over an MCM-22 zeolite (*Nature Commun.* **2018**, 9, 574, cited as ref 43), which may reflect the influence of the micropore network on hindering the migration of Pt species. These aspects are now discussed in the manuscript.

Action: Manuscript+ESI amended

3) In the title of this manuscript, the authors claim that, the redispersion of Pt nanoparticles into single atoms occur on "non-defect" support. I agree that, compared to CeO₂ or TiO₂, Fe₂O₃ may have less defects, such as oxygen vacancies. However, there are still other types of defects on the surface of Fe₂O₃ nanoparticles, due to the roughness of the surface (see: *Journal of Catalysis* 324 (2015) 127–132. *Journal of Electron Microscopy*, 58 (2009) 199–212.). Those unsaturated sites can be anchoring sites for Pt atoms. According to the data presented in this manuscript, the positions of Pt atoms on Fe₂O₃ cannot be determined. Therefore, I will suggest the authors not to use "non-defect" to describe the support and the corresponding redispersion behaviour unless they can provide more proof or more discussion on that point.

Response: The reviewer raises an important consideration. We must clarify that our title “Non defect-stabilized,” is not intended to suggest that our supports are defect-free, but rather that any defects present do not play a significant role in stabilizing single-atoms. This assertion is made on the basis that the defect concentration is far lower than the concentration of single-atoms. We agree that our as-synthesized Fe₂O₃ will contain some defects; however, after 800 °C calcination the intrinsic defect concentration is exceedingly small ($\sim 10^{-11}$ level)⁷² far below our Pt metal loading level (~ 1 %). If Pt single-atoms were solely stabilized by defects, then the maximum metal loading at which a SAC could be prepared would be extremely small (many orders of magnitude lower than we experimentally observe). Hence in our system, Pt single-atoms are not associated with defects. In any event, we have attempted to quantify the oxygen anion vacancy

concentration (the dominant form of surface defect in reducible metal oxides) by measuring the Lewis acidity of our Fe₂O₃-C800 support (without any Pt) by NH₃ chemisorption and subsequent TPD. Negligible chemisorbed amine was detectable confirming that our calcined Fe₂O₃ possessed **an extremely low surface defect concentration**, consistent with expectations. These additional experiments are now described in the manuscript.

The discovery that single-atom formation can be decoupled from a requirement for surface defects is highly significant, and underpins this work, and hence our preference is to retain the existing title. The fourth sentence of the Abstract clarifies any possible ambiguity “*Here we report that isolated Pt atoms can be stabilized through a strong metal-support interaction that is not associated with support defects*”.

Action: Manuscript+ESI amended

4) The in situ TEM experiments have been carried out to investigate the redispersion behaviour of Pt nanoparticles on Fe₂O₃, as presented in Figure 2. Please, make a careful look into the images, it seems that some of the nanoparticles (especially in the middle area) slightly grow. Does it mean the Ostwald ripening mechanism also occurs? The size distributions of the Pt nanoparticles should be provided for the fresh sample and the sample after 20 min at 800 °C to see the structural evolution.

Response: The reviewer raises an interesting question and suggestion. The size distribution of Pt nanoparticles is now added to Figure 3C and F by measuring every particle in Figure 3A and 3D according to this suggestion. Although two NPs in these images grew slightly after heating at 800 °C for 20 min, the majority shrank or even vanished. Consequently, the total particle number dropped from 300 to 200, and the average particle size decreased by 0.3 nm (from 3.0 nm to 2.7 nm). These changes were indeed accompanied by the growth of a few particles, suggesting that Ostwald ripening may compete with particle dispersion. This is unsurprising since migrating molecular PtO₂ species can be either trapped by the support or through encounters with large Pt NPs. However, prolonged heating resulted in the complete dispersion of all Pt NPs as single-atoms, evident in ex situ TEM images (Figure S8A and B) and from EXAFS (Figure 2, Figure S5, and Table S1).

Figure 3. In situ characterization of Pt NP oxidative dispersion.

Action: Manuscript amended

5) According to the experimental details for preparation of FeOx and Fe₂O₃, it seems that, there is a residual amount of sodium in the final solid carrier, as reported in the recent publication from their group (Chem. Sci., 2017, 8, 5126-5131). Does it come from the Al₂O₃ support? Does sodium play any role in the stabilization of single Pt atoms during the redispersion process? Control experiments using Na-free Fe₂O₃ as the support are recommended.

Response: The reviewer raises an excellent question related to recent work from Prof. Flytzani-Stephanopoulos's group which show that alkali cations can stabilize surface hydroxyls and in turn stabilize metal (Au and Pt) single-atoms and subnanometer clusters.⁷³⁻⁷⁵ We have not identified such a role for alkali in our systems,⁷⁶ and do not believe that residual sodium (from the Na₂CO₃ precipitant) plays a role in stabilizing Pt atoms in the present case. However, to test our hypothesis, we have prepared an alkali-free Fe₂O₃(N) support using (NH₄)₂CO₃ as the

precipitant, and subsequently functionalized this with 0.2 wt.% H₂PtCl₆ (see Methods-Fe₂O₃(N), H₂PtCl₆/Fe₂O₃(N), and H₂PtCl₆/Fe₂O₃(N)-C800 for synthesis details). TEM images confirmed the presence of Pt NPs on the H₂PtCl₆/Fe₂O₃(N) sample (Figure S16A and B). After calcination at 800 °C in air for 5 h, Pt NPs were dispersed as Pt single-atoms (Figure S16C and D), confirming that the reducible Fe₂O₃ support, and not presence of Na⁺, is responsible for stabilizing Pt single-atoms. These results are now summarized in the manuscript.

However, the surface area of our alkali-free Fe₂O₃(N) support was very small, < 1 m²/g, and so only able to fully disperse a lower Pt loading (0.2 wt%) as single-atoms compared with the Fe₂O₃ prepared using a Na₂CO₃. Alkalis may therefore be beneficial in preventing structural collapse of the oxide support during high temperature calcination.

Action: Manuscript+ESI amended

References

- 1 Mooiman, M. B., Sole, K.C, Dinham, N. *Metal Sustainability*. (wiley, 2016).
- 2 N. M. Davey, a. R. J. S. the platinum metals in electronics. *platinum Metals Review* **29**, 2-11, (1985).
- 3 Saunders, K., Davies, D., Golunski, S., Johnston, P. & Plucinski, P. Making the most of precious metal nanoparticles in the purification of industrial wastewater by Catalytic Wet Air Oxidation. *Johnson Matthey Technology Review*, (2018).
- 4 Ercolano, G., Cavaliere, S., Rozière, J. & Jones, D. J. Recent developments in electrocatalyst design thriving noble metals in fuel cells. *Current Opinion in Electrochemistry* **9**, 271-277, (2018).
- 5 Qiao, B. *et al.* Ultrastable single-atom gold catalysts with strong covalent metal-support interaction (CMSI). *Nano Res.* **8**, 2913-2924, (2015).
- 6 bp statistical review of world energy. (2016).
- 7 Urfels, L., Gélin, P., Primet, M. & Tena, E. Complete oxidation of methane at low temperature over Pt catalysts supported on high surface area SnO₂. *Top. Catal.* **30**, 427-432, (2004).
- 8 DeRita, L. *et al.* Catalyst Architecture for Stable Single Atom Dispersion Enables Site-Specific Spectroscopic and Reactivity Measurements of CO Adsorbed to Pt Atoms, Oxidized Pt Clusters, and Metallic Pt Clusters on TiO₂. *J. Am. Chem. Soc.* **139**, 14150-14165, (2017).
- 9 Nie, L. *et al.* Activation of surface lattice oxygen in single-atom Pt/CeO₂ for low-temperature CO oxidation. *Science* **358**, 1419, (2017).
- 10 Wang, C. *et al.* Water-Mediated Mars–Van Krevelen Mechanism for CO Oxidation on Ceria-Supported Single-Atom Pt₁ Catalyst. *ACS Catal.* **7**, 887-891, (2017).
- 11 Abbet, S. *et al.* Acetylene Cyclotrimerization on Supported Size-Selected Pd_n Clusters (1 ≤ n ≤ 30): One Atom Is Enough! *J. Am. Chem. Soc.* **122**, 3453-3457, (2000).
- 12 Kwak, J. H. *et al.* Coordinatively Unsaturated Al³⁺ Centers as Binding Sites for Active Catalyst Phases of Platinum on γ-Al₂O₃. *Science* **325**, 1670-1673, (2009).

- 13 Hackett, S. F. J. *et al.* High-Activity, Single-Site Mesoporous Pd/Al₂O₃ Catalysts for Selective Aerobic Oxidation of Allylic Alcohols. *Angew. Chem., Int. Ed.* **46**, 8593-8596, (2007).
- 14 Hu, B. *et al.* Isolated Fe^{II} on Silica As a Selective Propane Dehydrogenation Catalyst. *ACS Catal.* **5**, 3494-3503, (2015).
- 15 Wei, H. *et al.* Iced photochemical reduction to synthesize atomically dispersed metals by suppressing nanocrystal growth. *Nat. Commun.* **8**, 1490, (2017).
- 16 Wei, S. *et al.* Direct observation of noble metal nanoparticles transforming to thermally stable single atoms. *Nat. Nanotechnol.* **13**, 856-861, (2018).
- 17 Bulushev, D. A. *et al.* Single Atoms of Pt-Group Metals Stabilized by N-Doped Carbon Nanofibers for Efficient Hydrogen Production from Formic Acid. *ACS Catal.* **6**, 3442-3451, (2016).
- 18 Fei, H. *et al.* General synthesis and definitive structural identification of MN₄C₄ single-atom catalysts with tunable electrocatalytic activities. *Nat. Catal.* **1**, 63-72, (2018).
- 19 Bulushev, D. A. *et al.* Single Isolated Pd²⁺ Cations Supported on N-Doped Carbon as Active Sites for Hydrogen Production from Formic Acid Decomposition. *ACS Catal.* **6**, 681-691, (2016).
- 20 Zhang, L. *et al.* Coordination of Atomic Co-Pt Coupling Species at Carbon Defects as Active Sites for Oxygen Reduction Reaction. *J. Am. Chem. Soc.* **140**, 10757-10763, (2018).
- 21 Vile, G. *et al.* A stable single-site palladium catalyst for hydrogenations. *Angew. Chem. Int. Ed.* **54**, 11265-11269, (2015).
- 22 Chen, Z. *et al.* Stabilization of Single Metal Atoms on Graphitic Carbon Nitride. *Adv. Funct. Mater.* **27**, 1605785, (2017).
- 23 Yan, H. *et al.* Single-Atom Pd₁/Graphene Catalyst Achieved by Atomic Layer Deposition: Remarkable Performance in Selective Hydrogenation of 1,3-Butadiene. *J. Am. Chem. Soc.* **137**, 10484-10487, (2015).
- 24 Sun, S. *et al.* Single-atom Catalysis Using Pt/Graphene Achieved through Atomic Layer Deposition. *Sci. Rep.* **3**, 1775, (2013).
- 25 Wang, H. *et al.* Doping monolayer graphene with single atom substitutions. *Nano Lett.* **12**, 141-144, (2012).
- 26 Li, H. *et al.* Atomic Structure and Dynamics of Single Platinum Atom Interactions with Monolayer MoS₂. *ACS Nano* **11**, 3392-3403, (2017).
- 27 Liu, G. *et al.* MoS₂ monolayer catalyst doped with isolated Co atoms for the hydrodeoxygenation reaction. *Nat. Chem.* **9**, 810-816, (2017).
- 28 Jones, J. *et al.* Thermally stable single-atom platinum-on-ceria catalysts via atom trapping. *Science* **353**, 150-154, (2016).
- 29 Fu, Q., Deng, W., Saltsburg, H. & Flytzani-Stephanopoulos, M. Activity and stability of low-content gold-cerium oxide catalysts for the water-gas shift reaction. *Appl. Catal., B* **56**, 57-68, (2005).
- 30 Fu, Q., Saltsburg, H. & Flytzani-Stephanopoulos, M. Active Nonmetallic Au and Pt Species on Ceria-Based Water-Gas Shift Catalysts. *Science* **301**, 935-938, (2003).
- 31 Qiao, B. *et al.* Highly Efficient Catalysis of Preferential Oxidation of CO in H₂-Rich Stream by Gold Single-Atom Catalysts. *ACS Catal.* **5**, 6249-6254, (2015).
- 32 Dvorak, F. *et al.* Creating single-atom Pt-ceria catalysts by surface step decoration. *Nat. Commun.* **7**, 10801, (2016).
- 33 Bruix, A. *et al.* Maximum noble-metal efficiency in catalytic materials: atomically dispersed surface platinum. *Angew. Chem. Int. Ed.* **53**, 10525-10530, (2014).
- 34 Xie, P. *et al.* Nanoceria-Supported Single-Atom Platinum Catalysts for Direct Methane Conversion. *ACS Catal.*, 4044-4048, (2018).
- 35 Wei, H. *et al.* FeO_x-supported platinum single-atom and pseudo-single-atom catalysts for chemoselective hydrogenation of functionalized nitroarenes. *Nat. Commun.* **5**, 5634, (2014).

- 36 Gargallo-Caballero, R. *et al.* Co on Fe₃O₄(001): Towards precise control of surface properties. *J. Chem. Phys.* **144**, 094704, (2016).
- 37 Ryan, P. T. P. *et al.* Direct measurement of Ni incorporation into Fe₃O₄(001). *Phys Chem Chem Phys* **20**, 16469-16476, (2018).
- 38 Meier, M. *et al.* Probing the geometry of copper and silver adatoms on magnetite: quantitative experiment versus theory. *Nanoscale* **10**, 2226-2230, (2018).
- 39 Novotny, Z. *et al.* Ordered array of single adatoms with remarkable thermal stability: Au/Fe₃O₄(001). *Phys. Rev. Lett.* **108**, 216103, (2012).
- 40 Bliem, R. *et al.* Adsorption and incorporation of transition metals at the magnetite Fe₃O₄(001) surface. *Phys. Rev. B* **92**, 075440, (2015).
- 41 Zhang, S. *et al.* Catalysis on singly dispersed bimetallic sites. *Nat. Commun.* **6**, 7938, (2015).
- 42 Zhang, J. *et al.* Cation vacancy stabilization of single-atomic-site Pt₁/Ni(OH)_x catalyst for diboration of alkynes and alkenes. *Nat. Commun.* **9**, 1002, (2018).
- 43 Gu, X.-K. *et al.* Supported Single Pt₁/Au₁ Atoms for Methanol Steam Reforming. *ACS Catal.* **4**, 3886-3890, (2014).
- 44 Lang, R. *et al.* Hydroformylation of Olefins by a Rhodium Single-Atom Catalyst with Activity Comparable to RhCl(PPh₃)₃. *Angew. Chem. Int. Ed.* **55**, 16054-16058, (2016).
- 45 Wan, J. *et al.* Defect Effects on TiO₂ Nanosheets: Stabilizing Single Atomic Site Au and Promoting Catalytic Properties. *Adv. Mater.* **30**, (2018).
- 46 Sasahara, A., Pang, C. L. & Onishi, H. Probe Microscope Observation of Platinum Atoms Deposited on the TiO₂(110)-(1 × 1) Surface. *J. Phys. Chem. B* **110**, 13453-13457, (2006).
- 47 Liu, P. *et al.* Photochemical route for synthesizing atomically dispersed palladium catalysts. *Science* **352**, 797-800, (2016).
- 48 Wang, J. *et al.* Hydrogenolysis of Glycerol to 1,3-propanediol under Low Hydrogen Pressure over WO_x-Supported Single/Pseudo-Single Atom Pt Catalyst. *ChemSusChem* **9**, 784-790, (2016).
- 49 Liu, X., Duan, T., Meng, C. & Han, Y. Pt atoms stabilized on hexagonal boron nitride as efficient single-atom catalysts for CO oxidation: a first-principles investigation. *RSC Adv.* **5**, 10452-10459, (2015).
- 50 Lu, Z. *et al.* Pd₁/BN as a promising single atom catalyst of CO oxidation: a dispersion-corrected density functional theory study. *RSC Adv.* **5**, 84381-84388, (2015).
- 51 Ma, D. *et al.* CO catalytic oxidation on iron-embedded monolayer MoS₂. *Appl. Surf. Sci.* **328**, 71-77, (2015).
- 52 Ma, X. L., Liu, J. C., Xiao, H. & Li, J. Surface Single-Cluster Catalyst for N₂-to-NH₃ Thermal Conversion. *J. Am. Chem. Soc.* **140**, 46-49, (2018).
- 53 Tang, Y. *et al.* Adsorption behavior of Co anchored on graphene sheets toward NO, SO₂, NH₃, CO and HCN molecules. *Appl. Surf. Sci.* **342**, 191-199, (2015).
- 54 Liu, X. *et al.* Defect stabilized gold atoms on graphene as potential catalysts for ethylene epoxidation: a first-principles investigation. *Catal. Sci. Technol.* **6**, 1632-1641, (2016).
- 55 Li, F., Li, Y., Zeng, X. C. & Chen, Z. Exploration of High-Performance Single-Atom Catalysts on Support M₁/FeO_x for CO Oxidation via Computational Study. *ACS Catal.* **5**, 544-552, (2015).
- 56 Camellone, M. F. & Fabris, S. Reaction Mechanisms for the CO Oxidation on Au/CeO₂ Catalysts: Activity of Substitutional Au³⁺/Au⁺ Cations and Deactivation of Supported Au⁺ Adatoms. *J. Am. Chem. Soc.* **131**, 10473-10483, (2009).
- 57 Wang, Y.-G., Mei, D., Glezakou, V.-A., Li, J. & Rousseau, R. Dynamic formation of single-atom catalytic active sites on ceria-supported gold nanoparticles. *Nat. Commun.* **6**, 6511, (2015).
- 58 Wang, L. Q. *et al.* Preparation, characterization and catalytic performance of single-atom catalysts. *Chin. J. Catal.* **38**, 1528-1539, (2017).

- 59 Deng, T., Zheng, W. T. & Zhang, W. Increasing the range of non-noble-metal single-atom catalysts. *Chin. J. Catal.* **38**, 1489-1497, (2017).
- 60 Wang, Y., Zhang, W. H., Deng, D. H. & Bao, X. H. Two-dimensional materials confining single atoms for catalysis. *Chin. J. Catal.* **38**, 1443-1453, (2017).
- 61 Flytzani-Stephanopoulos, M. & Gates, B. C. Atomically Dispersed Supported Metal Catalysts. *Annu. Rev. Chem. Biomol. Eng.* **3**, 545-574, (2012).
- 62 Yang, X.-F. *et al.* Single-Atom Catalysts: A New Frontier in Heterogeneous Catalysis. *Acc. Chem. Res.* **46**, 1740-1748, (2013).
- 63 Liu, J. Catalysis by Supported Single Metal Atoms. *ACS Catal.*, 34-59, (2016).
- 64 Liang, S., Hao, C. & Shi, Y. The Power of Single-Atom Catalysis. *ChemCatChem* **7**, 2559-2567, (2015).
- 65 Gates, B. C., Flytzani-Stephanopoulos, M., Dixon, D. A. & Katz, A. Atomically dispersed supported metal catalysts: perspectives and suggestions for future research. *Catal. Sci. Technol.* **7**, 4259-4275, (2017).
- 66 Zhu, C., Fu, S., Shi, Q., Du, D. & Lin, Y. Single-Atom Electrocatalysts. *Angew. Chem. Int. Ed.* **56**, 13944-13960, (2017).
- 67 Liu, L. & Corma, A. Metal Catalysts for Heterogeneous Catalysis: From Single Atoms to Nanoclusters and Nanoparticles. *Chem. Rev.* **118**, 4981-5079, (2018).
- 68 Qin, R., Liu, P., Fu, G. & Zheng, N. Strategies for Stabilizing Atomically Dispersed Metal Catalysts. *Small Methods* **2**, 1700286, (2018).
- 69 Zhang, H., Liu, G., Shi, L. & Ye, J. Single-Atom Catalysts: Emerging Multifunctional Materials in Heterogeneous Catalysis. *Adv. Energy Mater.* **8**, 1701343, (2018).
- 70 Wang, A. Q., Li, J. & Zhang, T. Heterogeneous single-atom catalysis. *Nat. Rev. Chem.* **2**, 65-81, (2018).
- 71 Parkinson, G. S. Unravelling single atom catalysis: The surface science approach. *Chin. J. Catal.* **38**, 1454-1459, (2017).
- 72 Parkinson, G. S. Iron oxide surfaces. *Surface Science Reports* **71**, 272-365, (2016).
- 73 Zhai, Y. *et al.* Alkali-Stabilized Pt-OH_x Species Catalyze Low-Temperature Water-Gas Shift Reactions. *Science* **329**, 1633, (2010).
- 74 Yang, M. *et al.* Catalytically active Au-O(OH)_x-species stabilized by alkali ions on zeolites and mesoporous oxides. *Science* **346**, 1498, (2014).
- 75 Yang, M. *et al.* A Common Single-Site Pt(II)-O(OH)_x- Species Stabilized by Sodium on "Active" and "Inert" Supports Catalyzes the Water-Gas Shift Reaction. *J. Am. Chem. Soc.* **137**, 3470-3473, (2015).
- 76 Wei, H. *et al.* Remarkable effect of alkalis on the chemoselective hydrogenation of functionalized nitroarenes over high-loading Pt/FeO_x catalysts. *Chem. Sci.* **8**, 5126-5131, (2017).

Reviewers' Comments:

Reviewer #1:

Remarks to the Author:

The authors have addressed most of my issues. I do have some remaining concerns.

The response to the reviewer 1 on the importance of methane combustion needs to be in the Introduction as it makes the work more relevant.

The authors have added a large table with additional references to prior work in the SI. Although helpful, this is essentially unfair to the authors of the prior work as references in the SI do not receive citation counts. How does the journal handle this issue?

accept after addressing the methane combustion sentence.

In the future, it would be helpful in the response letter if the authors also put all of the changes in the text in the response letter to make it easier to see what they have changed.

Reviewer #2:

Remarks to the Author:

The authors have done lots of work to respond to reviewer comments. The EXAFS data are a good step forward. The work still lacks originality and the wording of the abstract (among others) does not stand up to scrutiny--an essential point is that the Pt is on various sites and they are not identified. This is just another paper showing atomically dispersed platinum on an oxide (there are others showing it on iron oxide) and a demonstration of catalysis of a reaction that has been shown before to occur on such catalysts. The work is publishable but does not belong in a journal that strives for forefront work.

Response to reviewers' comments

Reviewer #1 (Remarks to the Author):

The authors have addressed most of my issues. I do have some remaining concerns.

1. The response to the reviewer 1 on the importance of methane combustion needs to be in the Introduction as it makes the work more relevant.

Response: Thank you for your suggestion. The importance of methane combustion is now added in the Introduction: "In situ genesis of ferric oxide supported Pt SAC from Pt nanoparticles is verified in methane combustion reaction, one of the primary means of human energy production and important for mitigating environmental challenge associated with CH₄ emission."

Action: manuscript amended

2. The authors have added a large table with additional references to prior work in the SI. Although helpful, this is essentially unfair to the authors of the prior work as references in the SI do not receive citation counts. How does the journal handle this issue?

Response: Thank you for your suggestion. We are also very sorry for not citing all these elegant works in the manuscript due to reference limitations (< 70 references). So in the revised version, we added 6 more references in the Introduction, and we hope it will slightly compensate for the authors whose works have been cited in the SI.

The added references are:

[29] Fu, Q., Saltsburg, H. & Flytzani-Stephanopoulos, M. Active Nonmetallic Au and Pt Species on Ceria-Based Water-Gas Shift Catalysts. *Science* 301, 935-938 (2003).

[30] Fu, Q., Deng, W., Saltsburg, H. & Flytzani-Stephanopoulos, M. Activity and stability of low-content gold-cerium oxide catalysts for the water-gas shift reaction. *Appl. Catal., B* 56, 57-68 (2005).

[31] Hackett, S. F. J. et al. High-Activity, Single-Site Mesoporous Pd/Al₂O₃ Catalysts for Selective Aerobic Oxidation of Allylic Alcohols. *Angew. Chem., Int. Ed.* 46, 8593-8596 (2007).

[32] Sun, S. et al. Single-atom Catalysis Using Pt/Graphene Achieved through Atomic Layer Deposition. *Sci. Rep.* 3, 1775 (2013).

[33] Zhang, S. et al. Catalysis on singly dispersed bimetallic sites. *Nat. Commun.* 6, 7938, (2015).

[35] Liu, J. Catalysis by Supported Single Metal Atoms. *ACS Catal.* 7, 34-59 (2016).

Action: manuscript amended

3. accept after addressing the methane combustion sentence.

Response: Thank you for your suggestion. As discussed above in your first question, the sentence about the importance of methane combustion is now added in the Introduction.

Action: manuscript amended

4. In the future, it would be helpful in the response letter if the authors also put all of the changes in the text in the response letter to make it easier to see what they have changed.

Response: Thank you for your suggestion. We will follow your advice in the future.

Reviewer #2 (Remarks to the Author):

The authors have done lots of work to respond to reviewer comments. The EXAFS data are a good step forward. The work still lacks originality and the wording of the abstract (among others) does not stand up to scrutiny--an essential point is that the Pt is on various sites and they are not identified. This is just another paper showing atomically dispersed platinum on an oxide (there are others showing it on iron oxide) and a demonstration of catalysis of a reaction that has been shown before to occur on such catalysts. The work is publishable but does not belong in a journal that strives for forefront work.

Response: We thank the reviewer very much for all the suggestions which essentially improve our manuscript, although we respectfully disagree with some of the points in the above comments. We acknowledge that Pt atoms reside on various sites and they are not well identified. The identification of these sites on the one hand is extremely difficult in a practical catalyst system, if not impossible; on the other hand, it is not the focus of this work. We agree with the reviewer that Pt and other noble-metal atoms have been successfully dispersed on various metal-oxides, including ferric oxide. However in most cases, the electronic and/or structural defects associated with coordinatively unsaturated sites are used to stabilize metal atoms. As it is extremely difficult to create a high density of thermally stable defects on metal-oxide supports, the fabrication of high metal loading SACs with good thermal stability remains a great challenge so far. We demonstrate the utilization of metal-support interaction can be another way to stabilize SACs with high loading. The metal oxide reducibility dictates the ability of a support to anchor isolated Pt atoms. As a result, the non defect-stabilization strategy can be easily extended to non-reducible oxide by simply doping with iron oxide. This point has never been raised yet, so our work is quite original and provides deeper understanding into the origin of the thermal-stability for high-metal-loading SACs.

To our limited knowledge, SACs although successfully used in the non-oxidative coupling of methane (Science 344, 616-619, (2014), and ACS Catal., 8, 4044-4048, (2018).) and selective oxidation of methane to methanol or acetic acid (Nature 551, 605-608, (2017).), none of them has ever been applied in the methane combustion reaction which is very important in eliminating environmental concern. Moreover, we also observed the fascinating in situ genesis of Pt SAC from Pt nanoparticles during reaction, which clearly illustrated the dramatically enhanced catalytic performance. The Pt SAC is sinter-resistant at a temperature as high as 800 °C, which demonstrated the importance of strong metal-support interaction for the stabilization of SAC. We therefore believe our remarkable work meets the publication standard of Nature Communications, one of the most leading journals.